# DIFFERENTIALLY PRIVATE LEARNING NEEDS BETTER FEATURES (OR MUCH MORE DATA)

**Florian Tramèr**
Stanford University
`tramer@cs.stanford.edu`

**Dan Boneh**
Stanford University
`dabo@cs.stanford.edu`

## ABSTRACT

We demonstrate that differentially private machine learning has not yet reached its "AlexNet moment" on many canonical vision tasks: linear models trained on handcrafted features significantly outperform end-to-end deep neural networks for moderate privacy budgets. To exceed the performance of handcrafted features, we show that private learning requires either much more private data, or access to features learned on public data from a similar domain. Our work introduces simple yet strong baselines for differentially private learning that can inform the evaluation of future progress in this area.

## 1 INTRODUCTION

Machine learning (ML) models have been successfully applied to the analysis of sensitive user data such as medical images (Lundervold & Lundervold, 2019), text messages (Chen et al., 2019) or social media posts (Wu et al., 2016). Training these ML models under the framework of *differential privacy* (DP) (Dwork et al., 2006b; Chaudhuri et al., 2011; Shokri & Shmatikov, 2015; Abadi et al., 2016) can protect deployed classifiers against unintentional leakage of private training data (Shokri et al., 2017; Song et al., 2017; Carlini et al., 2019; 2020).

Yet, training deep neural networks with strong DP guarantees comes at a significant cost in utility (Abadi et al., 2016; Yu et al., 2020; Bagdasaryan et al., 2019; Feldman, 2020). In fact, on many ML benchmarks the reported accuracy of private deep learning still falls short of "shallow" (non-private) techniques. For example, on CIFAR-10, Papernot et al. (2020b) train a neural network to $66.2\%$ accuracy for a large DP budget of $\varepsilon = 7.53$, the highest accuracy we are aware of for this privacy budget. Yet, without privacy, higher accuracy is achievable with linear models and non-learned "handcrafted" features, e.g., (Coates & Ng, 2012; Oyallon & Mallat, 2015). This leads to the central question of our work:

*Can differentially private learning benefit from handcrafted features?*

We answer this question affirmatively by introducing simple and strong handcrafted baselines for differentially private learning, that significantly improve the privacy-utility guarantees on canonical vision benchmarks.

**Our contributions.**    We leverage the *Scattering Network* (ScatterNet) of Oyallon & Mallat (2015)—a *non-learned* SIFT-like feature extractor (Lowe, 1999)—to train linear models that improve upon the privacy-utility guarantees of deep learning on MNIST, Fashion-MNIST and CIFAR-10 (see Table 1). For example, on CIFAR-10 we exceed the accuracy reported by Papernot et al. (2020b) while simultaneously improving the provable DP-guarantee by $130\times$. On MNIST, we match the privacy-utility guarantees obtained with PATE (Papernot et al., 2018) *without requiring access to any public data*. We find that privately training deeper neural networks on handcrafted features also significantly improves over end-to-end deep learning, and even slightly exceeds the simpler linear models on CIFAR-10. Our results show that private deep learning remains outperformed by handcrafted priors on many tasks, and thus has yet to reach its "*AlexNet moment*" (Krizhevsky et al., 2012).

We find that models with handcrafted features outperform end-to-end deep models, *despite having more trainable parameters*. This is counter-intuitive, as the guarantees of private learning degrade

Table 1: Test accuracy of models with handcrafted ScatterNet features compared to prior results with end-to-end CNNs for various DP budgets ($\varepsilon, \delta = 10^{-5}$). Lower $\varepsilon$ values provide stronger privacy. The end-to-end CNNs with maximal accuracy for each privacy budget are underlined. We select the best ScatterNet model for each DP budget $\varepsilon \leq 3$ with a hyper-parameter search, and show the mean and standard deviation in accuracy for five runs.

| Data | $\varepsilon$-DP | Source | Test Accuracy (%) | | |
| --- | --- | --- | --- | --- | --- |
| | | | CNN | ScatterNet+linear | ScatterNet+CNN |
| MNIST | 1.2 | Feldman & Zrnic (2020) | 96.6 | $\mathbf{98.1 \pm 0.1}$ | $97.8 \pm 0.1$ |
| | 2.0 | Abadi et al. (2016) | 95.0 | $\mathbf{98.5 \pm 0.0}$ | $\mathbf{98.4 \pm 0.1}$ |
| | 2.32 | Bu et al. (2019) | 96.6 | $\mathbf{98.6 \pm 0.0}$ | $98.5 \pm 0.0$ |
| | 2.5 | Chen & Lee (2020) | 90.0 | $\mathbf{98.7 \pm 0.0}$ | $98.6 \pm 0.0$ |
| | 2.93 | Papernot et al. (2020a) | 98.1 | $\mathbf{98.7 \pm 0.0}$ | $\mathbf{98.7 \pm 0.1}$ |
| | 3.2 | Nasr et al. (2020) | 96.1 | – | |
| | 6.78 | Yu et al. (2019b) | 93.2 | – | |
| Fashion-MNIST | 2.7 | Papernot et al. (2020a) | 86.1 | $\mathbf{89.5 \pm 0.0}$ | $88.7 \pm 0.1$ |
| | 3.0 | Chen & Lee (2020) | 82.3 | $\mathbf{89.7 \pm 0.0}$ | $89.0 \pm 0.1$ |
| CIFAR-10 | 3.0 | Nasr et al. (2020) | 55.0 | $67.0 \pm 0.1$ | $\mathbf{69.3 \pm 0.2}$ |
| | 6.78 | Yu et al. (2019b) | 44.3 | – | – |
| | 7.53 | Papernot et al. (2020a) | 66.2 | – | – |
| | 8.0 | Chen & Lee (2020) | 53.0 | – | – |

with dimensionality in the worst case (Bassily et al., 2014).[1] We explain the benefits of handcrafted features by analyzing the convergence rate of *non-private* gradient descent. First, we observe that with low enough learning rates, training converges similarly with or without privacy (both for models with and without handcrafted features). Second, we show that handcrafted features significantly boost the convergence rate of *non-private* learning at low learning rates. As a result, when training with privacy, handcrafted features lead to more accurate models for a fixed privacy budget.

Considering these results, we ask: *what is the cost of private learning's "AlexNet moment"?* That is, which additional resources do we need in order to outperform our private handcrafted baselines? Following McMahan et al. (2018), we first consider the *data complexity* of private end-to-end learning. On CIFAR-10, we use an additional 500,000 labeled Tiny Images from Carmon et al. (2019) to show that about *an order of magnitude* more private training data is needed for end-to-end deep models to outperform our handcrafted features baselines. The high sample-complexity of private deep learning could be detrimental for tasks that cannot leverage "internet-scale" data collection (e.g., most medical applications).

We further consider private learning with access to *public* data from a similar domain. In this setting, handcrafted features can be replaced by features learned from public data via *transfer learning* (Razavian et al., 2014). While differentially private transfer learning has been studied in prior work (Abadi et al., 2016; Papernot et al., 2020a), we find that its privacy-utility guarantees have been underestimated. We revisit these results and show that with transfer learning, strong privacy comes at only a minor cost in accuracy. For example, given public *unlabeled* ImageNet data, we train a CIFAR-10 model to $92.7\%$ accuracy for a DP budget of $\varepsilon = 2$.

Our work demonstrates that higher quality features—whether handcrafted or transferred from public data—are of paramount importance for improving the performance of private classifiers in low (private) data regimes.

Code to reproduce our experiments is available at https://github.com/ftramer/Handcrafted-DP.

## 2   STRONG SHALLOW BASELINES FOR DIFFERENTIALLY PRIVATE LEARNING

We consider the standard *central* model of differential privacy (DP): a trusted party trains an ML model $f$ on a private dataset $D \in \mathcal{D}$, and publicly releases the model. The learning algorithm $A$

---

[1]A number of recent works have attempted to circumvent this worst-case dimensionality dependence by leveraging the empirical observation that model gradients lie in a low-dimensional subspace (Kairouz et al., 2020; Zhou et al., 2020b).

satisfies $(\varepsilon, \delta)$-differential privacy (Dwork et al., 2006a), if for any datasets $D, D'$ that differ in one record, and any set of models $S$:

$$\Pr[A(D) \in S] \leq e^{\varepsilon} \Pr[A(D') \in S] + \delta.$$

DP bounds an adversary's ability to infer information about any individual training point from the model. Cryptography can split the trust in a central party across users (Jayaraman et al., 2018; Bonawitz et al., 2017).

Prior work has trained private deep neural networks "end-to-end" (e.g., from image pixels), with large losses in utility (Shokri & Shmatikov, 2015; Abadi et al., 2016; Papernot et al., 2020b). In contrast, we study the benefits of handcrafted features that encode *priors* on the learning task's public domain (e.g., edge detectors for images). Although end-to-end neural networks outperform such features in the non-private setting, our thesis is that handcrafted features result in an easier learning task that is more amenable to privacy. We focus on computer vision, a canonical domain for private deep learning (Abadi et al., 2016; Yu et al., 2019b; Papernot et al., 2020b; Nasr et al., 2020)), with a rich literature on handcrafted features (Lowe, 1999; Dalal & Triggs, 2005; Bruna & Mallat, 2013). Our approach can be extended to handcrafted features in other domains, e.g., text or speech.

## 2.1 SCATTERING NETWORKS

We use the Scattering Network (ScatterNet) of Oyallon & Mallat (2015), a feature extractor that encodes natural image priors (e.g., invariance to small rotations and translations) using a cascade of wavelet transforms (Bruna & Mallat, 2013). As this cascade of transforms is data independent, we can obtain a differentially private classifier by privately fine-tuning a (linear) model on top of locally extracted features. In Appendix A, we discuss other candidate "non-deep" approaches that we believe to be less suitable for differentially private learning.

We use the default parameters in (Oyallon & Mallat, 2015), a ScatterNet $S(\boldsymbol{x})$ of depth two with wavelets rotated along eight angles. For images of size $H \times W$, this network extracts features of dimension $(K, {}^H/_4, {}^W/_4)$, with $K = 81$ for grayscale images, and $K = 243$ for RGB images. Note that the transform is thus *expansive*. More details on ScatterNets are in Appendix C.1.

## 2.2 DIFFERENTIALLY PRIVATE SCATTERNET CLASSIFIERS

To train private classifiers, we use the DP-SGD algorithm[2] of Abadi et al. (2016) (see Appendix B). DP-SGD works as follows: (1) batches of expected size $B$ are sampled at random;[3] (2) gradients are *clipped* to norm $C$; (3) Gaussian noise of variance $\sigma^2 C^2/B^2$ is added to the mean gradient. DP-SGD guarantees privacy for *gradients*, and is thus oblivious to preprocessing applied independently to each data sample, such as the ScatterNet transform.

When training a supervised classifier on top of ScatterNet features with gradient descent, we find that *normalizing* the features is crucial to obtain strong performance. We consider two approaches:

- *Group Normalization* (Wu & He, 2018): the channels of $S(\boldsymbol{x})$ are split into $G$ groups, and each is normalized to zero mean and unit variance. Data points are normalized independently so this step incurs no privacy cost.
- *Data Normalization*: the channels of $S(\boldsymbol{x})$ are normalized by their mean and variance across the training data. This step incurs a privacy cost as the per-channel means and variances need to be privately estimated.

Table 2 shows that normalization significantly accelerates convergence of *non-private* linear models trained on ScatterNet features, for MNIST, Fashion-MNIST and CIFAR-10. For CIFAR-10, Data

---

[2]Yu et al. (2019a) show that DP-SGD outperforms other algorithms for private convex optimization, e.g., logistic regression with output or objective perturbation (Chaudhuri et al., 2011; Bassily et al., 2014; Kifer et al., 2012). In Appendix D.3, we show that DP-SGD also outperforms *Privacy Amplification by Iteration* (Feldman et al., 2018) in our setting.

[3]Existing DP-SGD implementations (tensorflow/privacy, 2019; pytorch/opacus, 2020) and many prior works (e.g., (Abadi et al., 2016; Papernot et al., 2020b)) heuristically split the data into random batches of size *exactly* $B$. We use the same heuristic and show in Appendix D.4 that using the correct batch sampling does not affect our results.

Table 2: Effect of feature normalization on the test accuracy of *non-private* ScatterNet models after 20 epochs. We also report the maximal test accuracy upon convergence (mean and standard deviation over five runs).

| Dataset | Normalization (Test accuracy after 20 epochs) | | | |
|---|---|---|---|---|
| | None | Group Normalization | Data Normalization | Maximal Accuracy |
| MNIST | $95.9 \pm 0.0$ | $\mathbf{99.1 \pm 0.0}$ | $\mathbf{99.1 \pm 0.0}$ | $99.3 \pm 0.0$ |
| Fashion-MNIST | $82.6 \pm 0.1$ | $\mathbf{90.9 \pm 0.1}$ | $\mathbf{91.0 \pm 0.2}$ | $91.5 \pm 0.0$ |
| CIFAR-10 | $58.0 \pm 0.1$ | $67.8 \pm 0.2$ | $\mathbf{70.7 \pm 0.1}$ | $71.1 \pm 0.0$ |

Normalization performs significantly better than Group Normalization, so the small privacy cost of estimating channel statistics is warranted. While the maximal test accuracy of these models falls short of state-of-the-art CNNs, it exceeds all previously reported results for differentially private neural networks (even for large privacy budgets).

## 3  EVALUATING PRIVATE SCATTERNET CLASSIFIERS

We compare differentially private ScatterNet classifiers and deep learning models on MNIST (LeCun et al., 2010), Fashion-MNIST (Xiao et al., 2017) and CIFAR-10 (Krizhevsky, 2009). Many prior works have reported improvements over the DP-SGD procedure of Abadi et al. (2016) for these datasets. As we will show, ScatterNet classifiers outperform all prior approaches *while making no algorithmic changes to DP-SGD*. ScatterNet classifiers can thus serve as a strong canonical baseline for evaluating proposed improvements over DP-SGD in the future.

### 3.1  EXPERIMENTAL SETUP

Most prior works find the best model for a given DP budget using a hyper-parameter search. As the private training data is re-used many times, this overestimates the privacy guarantees. Private hyper-parameter search is possible at a small cost in the DP budget (Liu & Talwar, 2019), but we argue that fully accounting for this privacy leakage is hard as even our choices of architectures, optimizers, hyper-parameter *ranges*, etc. are informed by prior analysis of the same data. As in prior work, we thus do not account for this privacy leakage, and instead compare ScatterNet models and end-to-end CNNs with similar hyper-parameter searches. Moreover, we find that ScatterNet models are very robust to hyper-parameter changes and achieve near-optimal utility with random hyper-parameters (see Table 3). To evaluate ScatterNet models, we apply the following hyper-parameter search:

- We begin by fixing a *privacy schedule*. We target a moderate differential privacy budget of $(\varepsilon = 3, \delta = 10^{-5})$ and compute the noise scale $\sigma$ of DP-SGD so that the privacy budget is consumed after $T$ epochs. We try different values of $T$, with larger values resulting in training for more steps but with higher noise.
- We fix the gradient clipping threshold for DP-SGD to $C = 0.1$ for all our experiments. Thakkar et al. (2019) suggest to vary this threshold adaptively, but we did not observe better performance by doing so.
- We try various batch sizes $B$ and base learning rates $\eta$, with linear learning rate scaling (Goyal et al., 2017).[4]
- We try both Group Normalization (Wu & He, 2018) with different choices for the number of groups, and private Data Normalization with different choices of privacy budgets (see Appendix B for details).

We perform a grid-search over all parameters as detailed in Appendix C.5. We compare our ScatterNet classifiers to the CNN models of Papernot et al. (2020b) (see Appendix C.2), which achieve the

---

[4]Our decision to try various batch sizes is inspired by Abadi et al. (2016) who found that this parameter has a large effect on the performance of DP-SGD. Yet, in Appendix D.1 we show empirically, and argue formally that with a *linear learning rate scaling* (Goyal et al., 2017), DP-SGD performs similarly for a range of batch sizes. As a result, we recommend following the standard approach for tuning non-private SGD, wherein we fix the batch size and tune the learning rate.

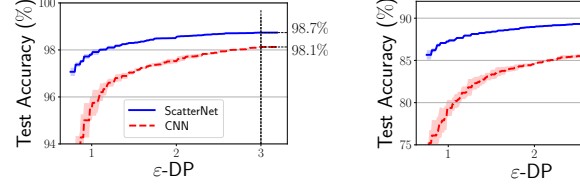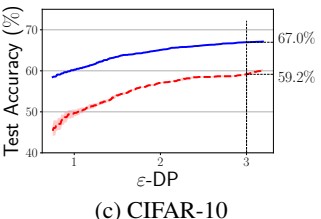

| (a) MNIST | (b) Fashion-MNIST | (c) CIFAR-10 |

Figure 1: Highest test accuracy achieved for each DP budget $(\varepsilon, \delta = 10^{-5})$ for ScatterNet classifiers and the end-to-end CNNs of Papernot et al. (2020b). We plot the mean and standard deviation across five runs.

highest reported accuracy for our targeted privacy budget for all three datasets. We also perform a grid-search for these models, which reproduces the results of Papernot et al. (2020b). We use the ScatterNet implementation from `Kymatio` (Andreux et al., 2020), and the DP-SGD implementation in `opacus` (pytorch/opacus, 2020) (formerly called `pytorch-dp`).

We use a NVIDIA Titan Xp GPU with 12GB of RAM for all our experiments. To run DP-SGD with large batch sizes $B$, we use the "virtual batch" approach of `opacus`: the average of clipped gradients is accumulated over multiple "mini-batches"; once $B$ gradients have been averaged, we add noise and take a gradient update step. Code to reproduce our experiments is available at `https://github.com/ftramer/Handcrafted-DP`.

### 3.2  RESULTS

To measure a classifier's accuracy for a range of privacy budgets, we compute the test accuracy as well as the DP budget $\varepsilon$ after each training epoch (with the last epoch corresponding to $\varepsilon = 3$). For various DP budgets $(\varepsilon, \delta = 10^{-5})$ used in prior work, Table 1 shows the maximal test accuracy achieved by a linear ScatterNet model in our hyper-parameter search, averaged over five runs. We also report results with CNNs trained on ScatterNet models, which are described in more detail below. Figure 1 further compares the full privacy-accuracy curves of our ScatterNets and of the CNNs of Papernot et al. (2020b). Linear models with handcrafted features significantly outperform prior results with end-to-end CNNs, for all privacy budgets $\varepsilon \leq 3$ we consider. Even when prior work reports results for larger budgets, they do not exceed the accuracy of our baseline.

In particular, for CIFAR-10, we match the best CNN accuracy in (Papernot et al., 2020b)—namely 66.2% for a budget of $\varepsilon = 7.53$—with a much smaller budget of $\varepsilon = 2.6$. This is an improvement in the DP-guarantee of $e^{4.9} \approx 134$. On MNIST, we significantly improve upon CNN models, and match the results of PATE (Papernot et al., 2018), namely 98.5% accuracy at $\varepsilon = 1.97$, in a more restricted setting (PATE uses 5,000 *public* unlabeled MNIST digits). In Appendix C.5, we provide the hyper-parameters that result in the highest test accuracy for our target DP budget of $(\varepsilon = 3, \delta = 10^{-5})$. We did not consider larger privacy budgets for ScatterNet classifiers, as the accuracy we achieve at $\varepsilon = 3$ is close to the accuracy of non-private ScatterNet models (see Table 2).

As noted above, our models (and those of most prior work) are the result of a hyper-parameter search. While we do not account for the privacy cost of this search, Table 3 shows that an additional advantage of ScatterNet classifiers is an increased robustness to hyper-parameter changes. In particular, for CIFAR-10 the *worst* configuration for linear ScatterNet classifiers outperforms the *best* configuration for end-to-end CNNs. Moreover, on MNIST and Fashion-MNIST, the median accuracy of linear ScatterNet models outperforms the best end-to-end CNN.

**Training CNNs on Handcrafted Features.**   Since *linear* models trained on handcrafted features outperform the privacy-utility guarantees of deep models trained end-to-end, a natural question is whether training deeper models on these features achieves even better results. We repeat the above experiment with a similar CNN model trained on ScatterNet features (see Appendix C.2). The privacy-accuracy curves for these models are in Figure 2. We find that handcrafted features also improve the utility of private deep models, a phenomenon which we analyze and explain in Section 4. On CIFAR-10, the deeper ScatterNet models even slightly outperform the linear models, while for MNIST and Fashion-MNIST the linear models perform best. This can be explained by the fact that

Table 3: Variability across hyper-parameters. For each model, we report the minimum, maximum, median and median absolute deviation (MAD) in test accuracy (in %) achieved for a DP budget of $(\varepsilon = 3, \delta = 10^{-5})$. The maximum accuracy below may exceed those in Table 1 and Figure 1, which are averages of five runs. SN stands for ScatterNet.

| Model | MNIST | | | | Fashion-MNIST | | | | CIFAR-10 | | | |
| --- | --- | --- | --- | --- | --- | --- | --- | --- | --- | --- | --- | --- |
| | Min | Max | Median | MAD | Min | Max | Median | MAD | Min | Max | Median | MAD |
| SN + Linear | **96.8** | **98.8** | **98.4** | **0.2** | **85.3** | 89.8 | **88.7** | **0.5** | **59.5** | 67.0 | 65.4 | **0.9** |
| SN + CNN | 95.6 | **98.8** | 98.1 | 0.3 | 77.8 | 89.1 | 87.2 | 1.0 | 57.3 | **69.5** | **66.9** | 1.6 |
| CNN | 86.1 | 98.2 | 97.4 | 0.5 | 20.2 | 86.2 | 83.6 | 1.8 | 39.4 | 59.2 | 52.5 | 5.4 |

Figure 2: Highest test accuracy achieved for each DP budget $(\varepsilon, \delta = 10^{-5})$ for linear ScatterNet classifiers, CNNs on top of ScatterNet features, and end-to-end CNNs. Shows mean and standard deviation across five runs.

in the *non-private setting*, linear ScatterNet models achieve close to state-of-the-art accuracy on MNIST and Fashion-MNIST, and thus there is little room for improvement with deeper models (see Table 11). Table 3 further shows that ScatterNet CNNs are also less sensitive to hyper-parameters than end-to-end CNNs.

Note that on each dataset we consider, end-to-end CNNs can outperform ScatterNet models when trained *without privacy*. Thus, end-to-end CNNs trained with DP-SGD must eventually surpass ScatterNet models for large enough privacy budgets. But this currently requires settling for weak provable privacy guarantees. On CIFAR-10 for example, ScatterNet classifiers still outperform end-to-end CNNs for $\varepsilon = 7.53$ (Papernot et al., 2020b). While the analysis of DP-SGD might not be tight, Jagielski et al. (2020) suggest that the true $\varepsilon$ guarantee of DP-SGD is at most one order of magnitude smaller than the current analysis suggests. Thus, surpassing handcrafted features for small privacy budgets on CIFAR-10 may require improvements beyond a tighter analysis of DP-SGD.

## 4 How Do Handcrafted Features Help?

In this section, we analyze *why* private models with handcrafted features outperform end-to-end CNNs. We first consider the *dimensionality* of our models, but show that this does not explain the utility gap. Rather, we find that the higher accuracy of ScatterNet classifiers is due to their faster convergence rate *when trained without noise*.

**Smaller models are not easier to train privately.** The utility of private learning typically degrades as the model's dimensionality increases (Chaudhuri et al., 2011; Bassily et al., 2014). This is also the case with DP-SGD which adds Gaussian noise, of scale proportional to the gradients, to *each* model parameter. We thus expect smaller models to be easier to train privately. Yet, as we see from Table 4, for MNIST and Fashion-MNIST the linear ScatterNet model has *more* parameters than the CNNs. For CIFAR-10, the end-to-end CNN we used is larger, so we repeat the experiment from Section 3 with a CNN of comparable size to the ScatterNet classifiers (see Appendix D.5). This has a minor effect on the performance of the CNN. *Thus, the dimensionality of ScatterNet classifiers fails to explain their better performance.*

**Models with handcrafted features converge faster *without privacy*.** DP-SGD typically requires a smaller learning rate than noiseless (clipped) SGD, so that the added noise gets averaged out over small steps. We indeed find that the optimal learning rate when training with DP-SGD is an order of magnitude lower than the optimal learning rate for training without noise addition (with gradients clipped to the same norm in both cases).

Table 4: Number of trainable parameters of our models. For CIFAR-10, we consider two different end-to-end CNN architectures (see Appendix C.2), the smaller of which has approximately as many parameters as the linear ScatterNet model.

|  | MNIST & Fashion-MNIST | CIFAR-10 |
|---|---|---|
| ScatterNet+Linear | 40K | 155K |
| ScatterNet+CNN | 33K | 187K |
| CNN | 26K | 551K / 168K |

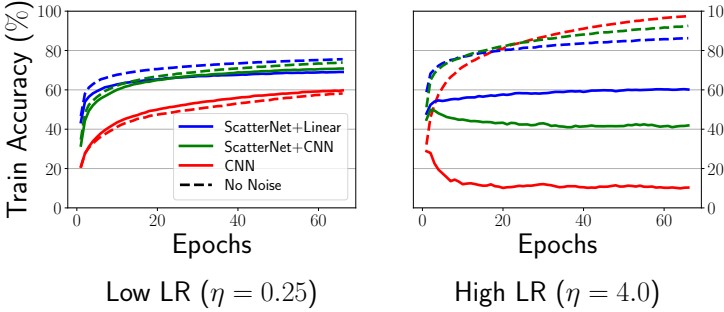

Figure 3: Convergence of DP-SGD with and without noise on CIFAR-10, for ScatterNet classifiers and end-to-end CNNs. (Left): low learning rate. (Right): high learning rate.

To understand the impact of gradient noise on the learning process, we conduct the following experiment: we select a low learning rate that is near-optimal for training models with gradient noise, and a high learning rate that is near-optimal for training without noise. For both learning rates, we train CIFAR-10 models both with and without noise (with gradient clipping in all cases). Figure 3 shows that with a high learning rate, all classifiers converge rapidly when trained without noise, but gradient noise vastly degrades performance. With a low learning rate however, training converges similarly whether we add noise or not. What distinguishes the ScatterNet models is the faster convergence rate of *noiseless* SGD. The experimental setup and similar qualitative results on MNIST and Fashion-MNIST are in Appendix C.6. Thus, we find that handcrafted features are beneficial for private learning because they result in a simpler learning task where training converges rapidly even with small update steps. Our analysis suggests two avenues towards obtaining higher accuracy with private deep learning:

- *Faster convergence:* Figure 3 suggests that faster convergence of *non-private* training could translate to better private learning. DP-SGD with adaptive updates (e.g., Adam (Kingma & Ba, 2015)) indeed sometimes leads to small improvements (Papernot et al., 2020b; Chen & Lee, 2020; Zhou et al., 2020a). Investigating private variants of *second-order optimization methods* is an interesting direction for future work.

- *More training steps (a.k.a more data):* For a fixed DP-budget $\varepsilon$ and noise scale $\sigma$, increasing the training set size $N$ allows for running more steps of DP-SGD (McMahan et al., 2018). In Section 5.1, we investigate how the collection of additional private data impacts the utility of private end-to-end models.

## 5 TOWARDS BETTER PRIVATE DEEP LEARNING

We have shown that on standard vision tasks, private learning strongly benefits from *handcrafted features*. Further improving our private baselines seems hard, as they come close to the maximal accuracy of ScatterNet models (see Table 2). We thus turn to other avenues for obtaining stronger privacy-utility guarantees. We focus on CIFAR-10, and discuss two natural paths towards better private models: (1) access to a larger *private* training set, and (2) access to a *public* image dataset from a different distribution (some works also consider access to public unlabeled data from the same distribution as the private data (Papernot et al., 2017; 2018; Zhu et al., 2020)).

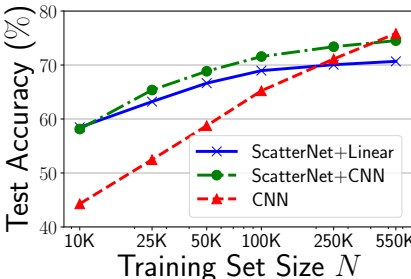
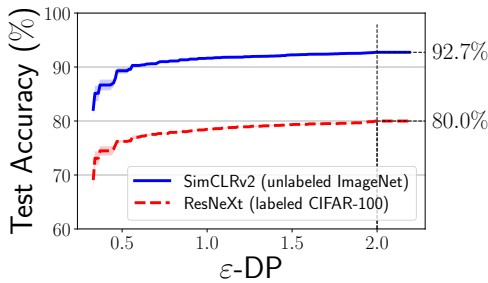

Figure 4: CIFAR-10 test accuracy for a training set of size $N$ and a DP budget of ($\varepsilon = 3, \delta = 1/2N$). For $N > 50$K, we augment CIFAR-10 with pseudo-labeled Tiny Images collected by Carmon et al. (2019).

Figure 5: Privacy-utility tradeoffs for transfer learning on CIFAR-10. We fine-tune linear models on features from a ResNeXt model trained on CIFAR-100, and from a SimCLR model trained on unlabeled ImageNet.

## 5.1 IMPROVING PRIVACY BY COLLECTING MORE DATA

We first analyze the benefits of additional *private labeled data* on the utility of private models. Since the privacy budget consumed by DP-SGD scales inversely with the size of the training data $N$, collecting more data allows either to train for more steps, or to lower the amount of noise added per step—for a fixed DP budget $\varepsilon$.

To obtain a larger dataset comparable to CIFAR-10, we use 500K pseudo-labeled Tiny Images[5] (Torralba et al., 2008) collected by Carmon et al. (2019).[6] We then train private models on subsets of size $10{,}000 \le N \le 550{,}000$ from this dataset. Figure 4 reports the highest test accuracy achieved for a privacy budget of ($\varepsilon = 3, \delta = 1/2N$) (see Appendix C.7 for the experimental setup). We find that we need about *an order-of-magnitude increase in the size of the private training dataset* in order for end-to-end CNNs to outperform ScatterNet features. As we show in Appendix C.7, larger datasets allow DP-SGD to be run for more steps at a fixed privacy budget and noise level (as also observed in (McMahan et al., 2018))—thereby overcoming the slow convergence rate we uncovered in Section 4. While the increased sample complexity of private deep learning might be viable for "internet-scale" applications (e.g., language modeling across mobile devices), it is detrimental for sensitive applications with more stringent data collection requirements, such as in healthcare.

## 5.2 TRANSFER LEARNING: BETTER FEATURES FROM PUBLIC DATA

Transfer learning is a natural candidate for privacy-preserving computer vision, as features learned on public image data often significantly outperform handcrafted features (Razavian et al., 2014). We first consider transfer learning from CIFAR-100 to CIFAR-10, where the labeled CIFAR-100 data is assumed public. We extract features from the penultimate layer of a ResNeXt (Xie et al., 2017) model trained on CIFAR-100. A non-private linear model trained on these features achieves 84% accuracy on CIFAR-10. When training linear models with DP-SGD, we get the privacy-utility curve in Figure 5 (see Appendix C.8 for details). We reach an accuracy of 80.0% at a budget of ($\varepsilon = 2, \delta = 10^{-5}$), a significant improvement over prior work for the same setting and privacy budget, e.g., 67% accuracy in (Abadi et al., 2016) and 72% accuracy in (Papernot et al., 2020a). The large gap between our results and prior work is mainly attributed to a better choice of *source model* (e.g., the transfer learning setup in (Papernot et al., 2020a) achieves 75% accuracy on CIFAR-10 *in the non-private setting*). Mirroring the work of Kornblith et al. (2019) on non-private transfer learning, we thus find that the heuristic rule "better models transfer better" also holds with differential privacy.

---

[5]The Tiny Images dataset has been withdrawn after the discovery of offensive class labels (Prabhu & Birhane, 2020). The subset used by Carmon et al. (2019) is filtered to match the CIFAR-10 labels, and is thus unlikely to contain offensive content.

[6]The privacy guarantees obtained with this dataset could be slightly overestimated, as the pseudo-labels of Carmon et al. (2019) are obtained using a model pre-trained on CIFAR-10, thus introducing dependencies between private data points.

We further consider access to a public dataset of *unlabeled* images. We extract features from the penultimate layer of a SimCLR model (Chen et al., 2020a) trained on unlabeled ImageNet. A non-private linear model trained on these features achieves $95\%$ accuracy on CIFAR-10 (using *labeled* ImageNet data marginally improves non-private transfer learning to CIFAR-10 (Chen et al., 2020a)). With the same setup as for CIFAR-100 (see Appendix C.8), we train a linear model to $92.7\%$ accuracy for a DP budget of $(\varepsilon = 2, \delta = 10^{-5})$ (see Figure 5).

## 6 CONCLUSION AND OPEN PROBLEMS

We have demonstrated that differentially private learning benefits from "handcrafted" features that encode priors on the learning task's domain. In particular, we have shown that private ScatterNet classifiers outperform end-to-end CNNs on MNIST, Fashion-MNIST and CIFAR-10. We have further found that handcrafted features can be surpassed when given access to more data, either a larger private training set, or a public dataset from a related domain. In addition to introducing strong baselines for evaluating future improvements to private deep learning and DP-SGD, our work suggests a number of open problems and directions for future work:

**Improving DP by accelerating convergence:** Our analysis in Section 4 shows that a limiting factor of private deep learning is the slow convergence rate of end-to-end deep models. While the existing literature on second-order optimization for deep learning has mainly focused on improving the overall *wall-clock time* of training, it suffices for DP to reduce the number of private *training steps*—possibly at an increase in computational cost.

**Federated learning:** While we have focused on a standard centralized setting for DP, our techniques can be extended to decentralized training schemes such as Federated Learning (McMahan et al., 2017; Bonawitz et al., 2017; Kairouz et al., 2019). DP has been considered for Federated Learning (Geyer et al., 2017; McMahan et al., 2018), but has also been found to significantly degrade performance in some settings (Yu et al., 2020).

**Handcrafted features for ImageNet and non-vision domains:** To our knowledge, there have not yet been any attempts to train ImageNet models with DP-SGD, partly due to the cost of computing per-sample gradients. While linear classifiers are unlikely to be competitive on ImageNet, handcrafted features can also help private learning by accelerating the convergence of CNNs, as we have shown in Figure 2. Notably, Oyallon et al. (2018) match the (non-private) accuracy of AlexNet (Krizhevsky et al., 2012) on ImageNet with a small six-layer CNN trained on ScatterNet features. Another interesting direction is to extend our results to domains beyond vision, e.g., with handcrafted features for text (Manning & Schutze, 1999) or speech (Andén & Mallat, 2014).

## ACKNOWLEDGEMENTS

We thank: Mani Malek, Ilya Mironov, Vaishaal Shankar and Ludwig Schmidt for fruitful discussions about differential privacy and computer vision baselines, and comments on early drafts of this paper; Nicolas Papernot and Shuang Song for helping us reproduce the results in (Papernot et al., 2020b); Nicolas Papernot for comments on early drafts of this paper; Edouard Oyallon for enlightening discussions about Scattering networks.

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

## A   WHY SCATTERNETS?

In this paper, we propose to use the ScatterNet features of Oyallon & Mallat (2015) as a basis for shallow differentially private vision classifiers. We briefly discuss a number of other shallow approaches that produce competitive results for canonical vision tasks, but which appear less suitable for private learning.

**Unsupervised feature dictionaries.**   Coates & Ng (2012) achieve above $80\%$ test accuracy on CIFAR-10 with linear models trained on top of a dictionary of features extracted from a mixture of image patches. Their approach relies on a combination of many 'tricks", including data normalization, data whitening, tweaks to standard Gaussian-Mixture-Model (GMM) algorithms, feature selection, etc. While it is conceivable that each of these steps could be made differentially private, we opt here for a much simpler unlearned baseline that is easier to analyze and to apply to a variety of different tasks. We note that existing work on differentially-private learning of mixtures (e.g., (Nissim et al., 2007)) has mainly focused on asymptotic guarantees, and we are not aware of any exiting algorithms that have been evaluated on high-dimensional datasets such as CIFAR-10.

**Kernel Machines.**   Recent work on *Neural Tangent Kernels* (Jacot et al., 2018) has shown that the performance of deep neural networks on CIFAR-10 could be matched by specialized kernel methods (Li et al., 2019; Arora et al., 2020; Shankar et al., 2020). Unfortunately, private learning with non-linear kernels is intractable in general (Chaudhuri et al., 2011; Rubinstein et al., 2012). Chaudhuri et al. (2011) propose to obtain private classifiers by approximating kernels using *random features* (Rahimi & Recht, 2008), but the very high dimensionality of the resulting learning problem makes it challenging to outperform our handcrafted features baseline. Indeed, we had originally considered a differentially-private variant of the random-feature CIFAR-10 classifier proposed in (Recht et al., 2018), but found the model's high dimensionality (over 10 million features) to be detrimental to private learning.

## B   DP-SGD, RDP AND PRIVATE DATA NORMALIZATION

Throughout this work, we use the DP-SGD algorithm of Abadi et al. (2016):

---
**Algorithm 1:** DP-SGD (Abadi et al., 2016)

---
**input** : Data $\{\boldsymbol{x}_1 \ldots, \boldsymbol{x}_N\}$, learning rate $\eta$, noise scale $\sigma$, batch size $B$, gradient norm bound $C$, epochs $T$

1   Initialize $\boldsymbol{\theta}_0$ randomly

   **for** $t \in [T \cdot N/B]$ **do**

2      Sample a batch $\boldsymbol{B}_t$ by selecting each $\boldsymbol{x}_i$ independently with probability $B/N$

3      For each $\boldsymbol{x}_i \in \boldsymbol{B}_t$: $\boldsymbol{g}_t(\boldsymbol{x}_i) \leftarrow \nabla_{\boldsymbol{\theta}_t} L(\boldsymbol{\theta}_t, \boldsymbol{x}_i)$     `// compute per-sample gradients`

4                $\tilde{\boldsymbol{g}}_t(\boldsymbol{x}_i) \leftarrow \boldsymbol{g}_t(\boldsymbol{x}_i) \cdot \min(1, C/\|\boldsymbol{g}_t(\boldsymbol{x}_i)\|_2)$     `// clip gradients`

5      $\tilde{\boldsymbol{g}}_t \leftarrow \frac{1}{B}\left(\sum_{\boldsymbol{x}_i \in \boldsymbol{B}_t} \tilde{\boldsymbol{g}}_t(\boldsymbol{x}_i) + \mathcal{N}(0, \sigma^2 C^2 \boldsymbol{I})\right)$ `// add noise to average gradient with`
     `Gaussian mechanism`

6      $\boldsymbol{\theta}_{t+1} \leftarrow \boldsymbol{\theta}_t - \eta \tilde{\boldsymbol{g}}_t$         `// SGD step`

**output** : $\boldsymbol{\theta}_{TN/B}$

---

The tightest known privacy analysis of the DP-SGD algorithm is based on the notion of Rényi differential privacy (RDP) from Mironov (2017), which we recall next.

**Definition B.1** (Rényi Divergence). For two probability distributions $P$ and $Q$ defined over a range $\mathcal{R}$, the Rényi divergence of order $\alpha > 1$ is

$$D_\alpha(P\|Q) \coloneqq \frac{1}{\alpha - 1} \log \mathop{\mathbb{E}}_{x \sim Q} \left( \frac{P(x)}{Q(x)} \right)^\alpha \;.$$

**Definition B.2** (($\alpha, \varepsilon$)-RDP (Mironov, 2017)). A randomized mechanism $f : \mathcal{D} \to \mathcal{R}$ is said to have $\varepsilon$-Rényi differential privacy of order $\alpha$, or ($\alpha, \varepsilon$)-RDP for short, if for any adjacent $D, D' \in \mathcal{D}$ it holds that

$$D_\alpha(f(D)\|f(D')) \leq \varepsilon \;.$$

To analyze the privacy guarantees of DP-SGD, we numerically compute $D_\alpha(f(D)\|f(D'))$ for a range of orders $\alpha$ (Mironov et al., 2019; Wang et al., 2019) in each training step, where $D$ and $D'$ are training sets that differ in a single element. To obtain privacy guarantees for $t$ training steps, we use the composition properties of RDP:

**Lemma B.3** (Adaptive composition of RDP (Mironov et al., 2019)). *Let $f : \mathcal{D} \to \mathcal{R}_1$ be ($\alpha, \varepsilon_1$)-RDP and $g : \mathcal{R}_1 \times \mathcal{D} \to \mathcal{R}_2$ be ($\alpha, \varepsilon_2$)-RDP, then the mechanism defined as $(X, Y)$, where $X \sim f(D)$ and $Y \sim g(X, D)$, satisfies ($\alpha, \varepsilon_1 + \varepsilon_2$)-RDP.*

Finally, the RDP guarantees of the full DP-SGD procedure can be converted into a ($\varepsilon, \delta$)-DP guarantee:

**Lemma B.4** (From RDP to ($\varepsilon, \delta$)-DP (Mironov et al., 2019)). *If $f$ is an ($\alpha, \varepsilon$)-RDP mechanism, it also satisfies $(\varepsilon + \frac{\log 1/\delta}{\alpha - 1}, \delta)$-DP for any $0 < \delta < 1$.*

**Private Data Normalization.** In order to apply Data Normalization to the ScatterNet features (which greatly improves convergence, especially on CIFAR-10), we use the `PrivDataNorm` procedure in Algorithm 2 to compute private estimates of the per-channel mean and variance of the ScatterNet features.

---

**Algorithm 2:** Private Data Normalization

---

**Function** `PrivChannelMean` (*data $\boldsymbol{D} \in \mathbb{R}^{N \times K \times H \times W}$, norm bound $C$, noise scale $\sigma_{norm}$*)

1    For $1 \leq i \leq N$: $\boldsymbol{\mu}_i \leftarrow \mathbb{E}_{h,w}\left[\boldsymbol{D}_{(i,\cdot,h,w)}\right] \in \mathbb{R}^K$     `// compute per-channel means for each sample`

2    $\boldsymbol{\mu}_i \leftarrow \boldsymbol{\mu}_i \cdot \min(1, {}^C/_{\|\boldsymbol{\mu}_i\|_2})$     `// clip each sample's per-channel means`

3    $\tilde{\boldsymbol{\mu}} \leftarrow \mathbb{E}_i[\boldsymbol{\mu}_i] + \frac{1}{N}\mathcal{N}(0, \sigma_{\text{norm}}^2 C^2 \boldsymbol{I})$     `// private mean using Gaussian mechanism`

4    **return** $\tilde{\mu}$

**Function** `PrivDataNorm` (*data $\boldsymbol{D}$, norm bounds $C_1, C_2$, noise scale $\sigma_{norm}$, threshold $\tau$*)

1    $\tilde{\boldsymbol{\mu}} \leftarrow$ `PrivChannelMean`$(\boldsymbol{D}, C_1, \sigma_{\text{norm}})$     `// private per-channel mean`

2    $\tilde{\boldsymbol{\mu}}_{\boldsymbol{D}^2} \leftarrow$ `PrivChannelMean`$(\boldsymbol{D}^2, C_2, \sigma_{\text{norm}})$     `// private per-channel mean-square`

3    $\tilde{\text{Var}} \leftarrow \max(\tilde{\boldsymbol{\mu}}_{\boldsymbol{D}^2} - \tilde{\boldsymbol{\mu}}^2, \tau)$     `// private per-channel variance`

4    For each $1 \leq i \leq N$, $\hat{\boldsymbol{D}}_i \leftarrow (\boldsymbol{D}_i - \tilde{\boldsymbol{\mu}})/\sqrt{\tilde{\text{Var}}}$     `// normalize each sample independently`

5    **return** $\hat{D}$

---

In order to obtain tight privacy guarantees for the full training procedure (i.e., privacy-preserving Data Normalization followed by DP-SGD), we first derive the RDP guarantees of `PrivDataNorm`:

**Claim B.5.** *The `PrivDataNorm` procedure is $(\alpha, \alpha/\sigma_{norm}^2)$-RDP for any $\alpha > 1$.*

The above claim follows from the RDP guarantees of the Gaussian mechanism in (Mironov, 2017), together with the composition properties of RDP in Lemma B.3 above.

Finally, given an RDP guarantee of ($\alpha, \varepsilon_1$) for `PrivDataNorm`, and an RDP guarantee of ($\alpha, \varepsilon_2$) for DP-SGD, we apply Lemma B.3 to obtain an RDP guarantee of ($\alpha, \varepsilon_1 + \varepsilon_2$), and convert to a DP guarantee using Lemma B.4.

## C   EXPERIMENTAL SETUP

### C.1   SCATTERING NETWORKS

We briefly review the scattering network (ScatterNet) of Oyallon & Mallat (2015). Consider an input $x$. The output of a scattering network of depth $J$ is a feature vector given by

$$S(\boldsymbol{x}) \coloneqq A_J \left| W_2 \left| W_1 \, \boldsymbol{x} \right| \right| , \tag{1}$$

where the operators $W_1$ and $W_2$ are complex-valued wavelet transforms, each followed by a non-linear complex modulus, and the final operator $A$ performs spatial averaging over patches of $2^J$ features. Both wavelet transforms $W_1$ and $W_2$ are linear operators that compute a cascade of convolutions with filters from a fixed family of wavelets. For an input image of spatial dimensions $H \times W$, the ScatterNet is applied to each of the image's color channels independently to yield an output tensor of dimension $(K, \frac{H}{2^J}, \frac{W}{2^J})$. The channel dimensionality $K$ depends on the network depth $J$ and the granularity of the wavelet filters, and is chosen so that $K/2^{2J} = O(1)$ (i.e., the ScatterNet approximately preserves the data dimensionality).

For all experiments, we use the default parameters proposed by Oyallon & Mallat (2015), namely a Scattering Network of depth $J = 2$, consisting of wavelet filters rotated along eight angles. For an an input image of spatial dimensions $H \times W$, this configuration produces an output of dimension $(K, H/4, W/4)$, with $K = 81$ for grayscale images, and $K = 243$ for RGB images.

### C.2   MODEL ARCHITECTURES

Below, we describe the ScatterNet+Linear, ScatterNet+CNN and end-to-end CNN architectures used in Section 3 and Section 4. The CNN architectures are adapted from Papernot et al. (2020b).

**Linear ScatterNet Classifiers.**   The default Scattering Network of Oyallon & Mallat (2015) extracts feature vectors of size $(81, 7, 7)$ for MNIST and Fashion-MNIST and of size $(243, 8, 8)$ for CIFAR-10. We then train a standard logistic regression classifier (with per-class bias) on top of these features, as summarized below:

Table 5: Size of linear ScatterNet classifiers.

| Dataset | Image size | Linear ScatterNet size |
|---|---|---|
| MNIST | $28 \times 28$ | $3969 \times 10$ |
| Fashion-MNIST | $28 \times 28$ | $3969 \times 10$ |
| CIFAR-10 | $32 \times 32 \times 3$ | $15552 \times 10$ |

**End-to-end CNNs.**   We use the CNN architectures proposed by Papernot et al. (2020b), which were found as a result of an architecture search tailored to DP-SGD.[7] Notably, these CNNs are quite small (since the noise of DP-SGD grows with the model's dimensionality) and use Tanh activations, which Papernot et al. (2020b) found to outperform the more common ReLU activations. For the experiments in Section 4, we also consider a smaller CIFAR-10 model, with a dimensionality comparable to the linear ScatterNet classifier. While the standard model has six convolutional layers of size 32-32-64-64-128-128, the smaller model has five convolutional layers of size 16-16-32-32-64 (with max-pooling after the 2nd, 4th and 5th convolution).

---

[7]The CNN architecture for CIFAR-10 in Table 7 differs slightly from that described in (Papernot et al., 2020b). Based on discussions with the authors of (Papernot et al., 2020b), the architecture in Table 7 is the correct one to reproduce their best results.

Table 6: End-to-end CNN model for MNIST and Fashion-MNIST, with Tanh activations (Papernot et al., 2020b).

| Layer | Parameters |
|---|---|
| Convolution | 16 filters of 8x8, stride 2, padding 2 |
| Max-Pooling | 2x2, stride 1 |
| Convolution | 32 filters of 4x4, stride 2, padding 0 |
| Max-Pooling | 2x2, stride 1 |
| Fully connected | 32 units |
| Fully connected | 10 units |

Table 7: End-to-end CNN model for CIFAR-10, with Tanh activations (Papernot et al., 2020b). In Section 4, we also use a smaller variant of this architecture with five convolutional layers of 16-16-32-32-64 filters.

| Layer | Parameters |
|---|---|
| Convolution x2 | 32 filters of 3x3, stride 1, padding 1 |
| Max-Pooling | 2x2, stride 2 |
| Convolution x2 | 64 filters of 3x3, stride 1, padding 1 |
| Max-Pooling | 2x2, stride 2 |
| Convolution x2 | 128 filters of 3x3, stride 1, padding 1 |
| Max-Pooling | 2x2, stride 2 |
| Fully connected | 128 units |
| Fully connected | 10 units |

**ScatterNet CNNs.** To fine-tune CNNs on top of ScatterNet features, we adapt the CNNs from Table 6 and Table 7. As the ScatterNet feature vector is larger than the input image ($784 \rightarrow 3969$ features for MNIST and Fashion-MNIST, and $3072 \rightarrow 15552$ features for CIFAR-10), we use smaller CNN models. For MNIST and Fashion MNIST, we reduce the number of convolutional filters. For CIFAR-10, we reduce the network depth from 8 to 3, which results in a model with approximately as many parameters as the linear ScatterNet classifier.

Table 8: CNN model fine-tuned on ScatterNet features for MNIST and Fashion-MNIST, with Tanh activations.

| Layer | Parameters |
|---|---|
| Convolution | 16 filters of 3x3, stride 2, padding 1 |
| Max-Pooling | 2x2, stride 1 |
| Convolution | 32 filters of 3x3, stride 1, padding 1 |
| Max-Pooling | 2x2, stride 1 |
| Fully connected | 32 units |
| Fully connected | 10 units |

Table 9: CNN model on ScatterNet features for CIFAR-10, with Tanh activations. In Section 4, we also use a smaller variant of this model with four convolutional layers of 16-16-32-32 filters.

| Layer | Parameters |
|---|---|
| Convolution | 64 filters of 3x3, stride 1, padding 1 |
| Max-Pooling | 2x2, stride 2 |
| Convolution | 64 filters of 3x3, stride 1, padding 1 |
| Max-Pooling | 2x2, stride 2 |
| Fully connected | 10 units |

## C.3 Effect of Normalization

To evaluate the effect of feature normalization in Table 2, we train linear models on ScatterNet features using DP-SGD without noise ($\sigma = 0$). We train one model without feature normalization, one with Data Normalization, and three with Group Normalization (Wu & He, 2018) with $G \in \{9, 27, 81\}$ groups. For Group Normalization, Table 2 reports results for the best choice of groups. The remaining hyper-parameters are given below.

Table 10: Hyper-parameters for evaluating the effect of feature normalization in Table 2.

| Parameter | MNIST | Fashion-MNIST | CIFAR-10 |
|---|---|---|---|
| Gradient clipping norm $C$ | 0.1 | 0.1 | 0.1 |
| Momentum | 0.9 | 0.9 | 0.9 |
| Epochs $T$ | 20 | 20 | 20 |
| Batch size $B$ | 512 | 512 | 512 |
| Learning rate $\eta$ | 2 | 4 | 2 |
| Best choice of groups $G$ | 27 | 81 | 27 |

## C.4 Non-Private Model Performance

For each of the model architectures described in Appendix C.2, we report the best achieved test accuracy without privacy, and without any other form of explicit regularization. For MNIST and Fashion-MNIST, fine-tuning a linear model or a CNN on top of ScatterNet features results in similar performance, whereas on CIFAR-10, the CNN performs slightly better. For Fashion-MNIST the end-to-end CNN performs slightly worse than the linear model (mainly due to a lack of regularization). For CIFAR-10, the end-to-end CNN significantly outperforms the ScatterNet models.

Table 11: Test accuracy (in %) for models trained without privacy. Average and standard deviation are computed over five runs.

| Dataset | ScatterNet+Linear | ScatterNet+CNN | CNN |
|---|---|---|---|
| MNIST | $99.3 \pm 0.0$ | $99.2 \pm 0.0$ | $99.2 \pm 0.0$ |
| Fashion-MNIST | $91.5 \pm 0.0$ | $91.5 \pm 0.2$ | $90.1 \pm 0.2$ |
| CIFAR-10 | $71.1 \pm 0.0$ | $73.8 \pm 0.3$ | $80.0 \pm 0.1$ |

## C.5 Evaluating Private ScatterNet Classifiers

We use DP-SGD with momentum for all experiments. Prior work found that the use of adaptive optimizers (e.g., Adam (Kingma & Ba, 2015)) provided only marginal benefits for private learning (Papernot et al., 2020a). Moreover, we use no data augmentation, weight decay, or other mechanisms aimed at preventing overfitting. The reason is that differential privacy is itself a powerful regularizer (informally, differential privacy implies low generalization error (Dwork et al., 2015)), so our models all *underfit* the training data.

The table below lists the ranges of hyper-parameters used for the experiments in Section 3, to train linear ScatterNet classifiers, end-to-end CNNs, and CNNs fine-tuned on ScatterNet features.

Table 12: Hyper-parameters for the evaluation of private linear classifiers fine-tuned on ScatterNet features, CNNs fine-tuned on ScatterNet features, and end-to-end CNNs in Section 3.

| Parameter | MNIST | Fashion-MNIST | CIFAR-10 |
|---|---|---|---|
| DP guarantee $(\varepsilon, \delta)$ | $(3, 10^{-5})$ | $(3, 10^{-5})$ | $(3, 10^{-5})$ |
| Gradient clipping norm $C$ | 0.1 | 0.1 | 0.1 |
| Momentum | 0.9 | 0.9 | 0.9 |
| Batch size $B$ | $\{512, 1024, \dots, 16384\}$ | $\{512, 1024, \dots, 16384\}$ | $\{512, 1024, \dots, 16384\}$ |
| Learning rate $\eta$ | $\{1/4, 1/2, 1, 2\} \cdot {}^{B}/_{512}$ | $\{1/4, 1/2, 1, 2\} \cdot {}^{B}/_{512}$ | $\{1/8, 1/4, 1/2, 1\} \cdot {}^{B}/_{512}$ |
| Epochs $T$ | $\{15, 25, 40\}$ | $\{15, 25, 40\}$ | $\{30, 60, 120\}$ |
| DP-SGD noise scale $\sigma$ | calculated numerically so that a DP budget of $(\varepsilon, \delta)$ is spent after $T$ epochs | | |
| Group Norm. groups $G$ | $\{9, 27, 81\}$ | $\{9, 27, 81\}$ | $\{9, 27, 81\}$ |
| Data Norm. $(C_1, C_2, \sigma_{\text{norm}})$ | $(0.2, 0.05, \{6, 8\})$ | $(0.3, 0.15, \{6, 8\})$ | $(1.0, 1.5, \{6, 8\})$ |

In Table 13, we give the set of hyper-parameters that resulted in the maximal accuracy for our target DP budget of $(\varepsilon = 3, \delta = 10^{-5})$. For each model, we report the *base* learning rate, before re-scaling by ${}^{B}/_{512}$. We find that some hyper-parameters that result in the best performance are at the boundary of our search range. Yet, as we show in Figure 8, modifying these hyper-parameters results in no significant upward trend, so we refrained from further increasing our search space.

Table 13: Set of hyper-parameters resulting in the highest test accuracy for a privacy budget of $(\varepsilon = 3, \delta = 10^{-5})$. Note that we report the *base* learning rate (LR), before scaling by a factor of ${}^{B}/_{512}$. SN stands for ScatterNet.

| Parameter | MNIST | | | Fashion-MNIST | | | CIFAR-10 | | |
|---|---|---|---|---|---|---|---|---|---|
| | SN+Linear | SN+CNN | CNN | SN+Linear | SN+CNN | CNN | SN+Linear | SN+CNN | CNN |
| Batch size $B$ | 4096 | 1024 | 512 | 8192 | 2048 | 2048 | 8192 | 8192 | 1024 |
| Base LR $\eta$ | 1 | 1/2 | 1/2 | 1 | 1 | 1 | 1/4 | 1/4 | 1/2 |
| Epochs $T$ | 40 | 25 | 40 | 40 | 40 | 40 | 60 | 60 | 30 |
| Groups $G$ | - | - | - | 27 | 27 | - | - | - | - |
| Data Norm. $\sigma_{\text{norm}}$ | 8 | 8 | - | - | - | - | 8 | 8 | - |

## C.6 MEASURING MODEL CONVERGENCE SPEED

For the experiments in Section 4, we compare the convergence of the models from Appendix C.2 when trained with and without noise, and with either a low or high learning rate. The table below lists the hyper-parameters for the CIFAR-10 experiment in Figure 3, as well as for the corresponding experiments for MNIST and Fashion-MNIST in Figure 11. When training without privacy, we still clip gradients to a maximal norm of $C = 0.1$, but omit the noise addition step of DP-SGD (and we also omit the noise when using Data Normalization).

Table 14: Hyper-parameters for the experiments on model convergence rates in Figure 3 and Figure 11.

| Dataset | Batch size $B$ | Gradient Norm $C$ | Learning rate $\eta$ (low, high) | Epochs $T$ | Normalization |
|---|---|---|---|---|---|
| MNIST | 512 | 0.1 | $(1/2, 8)$ | 40 | Data Norm. $(\sigma_{\text{norm}} = 8)$ |
| Fashion-MNIST | 512 | 0.1 | $(1, 16)$ | 40 | Group Norm. $(G = 81)$ |
| CIFAR-10 | 512 | 0.1 | $(1/4, 4)$ | 60 | Data Norm. $(\sigma_{\text{norm}} = 8)$ |

## C.7 PRIVATE LEARNING ON LARGER DATASETS

For the experiment in Section 5.1, we use an additional 500K images from the Tiny Images dataset (Torralba et al., 2008), which were collected and labeled by Carmon et al. (2019) using a pre-trained CIFAR-10 classifier (see (Carmon et al., 2019, Appendix B.6) for details on the selec-

tion process for this dataset).[8] We create datasets of size $N \in \{10\text{K}, 25\text{K}, 50\text{K}, 100\text{K}, 250\text{K}, 550\text{K}\}$ by taking subsets of this larger dataset. We only use the data of Carmon et al. (2019) to complement the CIFAR-10 dataset when $N > 50\text{K}$. As noted by Carmon et al. (2019), the additional 500K images do not entirely match the distribution of CIFAR-10. Nevertheless, we find that training our classifiers *without privacy* on augmented datasets of size $N > 50\text{K}$ does not negatively impact the test accuracy on CIFAR-10.

For each training set size, we re-train our models with a hyper-parameter search. To limit computational cost, and informed by our prior experiments, we fix some parameters, as shown in Table 15. When applying Data Normalization to ScatterNet features, we compute the per-channel statistics only over the original CIFAR-10 samples, and compute the privacy guarantees of `PrivDataNorm` using the Rényi DP analysis of the sampled Gaussian mechanism (Mironov et al., 2019; Wang et al., 2019).

Table 15: Hyper-parameters for the evaluation of private classifiers on larger datasets in Section 5.1.

| Parameter | Value for dataset of size $N$ |
|---|---|
| DP guarantee $(\varepsilon, \delta)$ | $(3, 1/2N)$ |
| Gradient norm $C$ | 0.1 |
| Momentum | 0.9 |
| Batch size $B$ | 8192 |
| Learning rate $\eta$ | $\{1/8, 1/4, 1/2, 1, 2\} \cdot 8192/512$ |
| Epochs $T$ | $\{15, 30, 60, 120\} \cdot 50000/N$ |
| Data Norm. params $(C_1, C_2, \sigma_{\text{norm}})$ | $(1, 1.5, 8)$ |

The only hyper-parameters are thus the number of epochs (normalized by the size of the original CIFAR-10 data) and the learning rate $\eta$. The optimal values we found for these parameters are given below in Table 16. As we increase the dataset size, we obtain better accuracy by training for more steps and with higher learning rates. Figure 4 reports the final accuracy for these best-performing models.

Table 16: Set of hyper-parameters resulting in the highest test accuracy for a privacy budget of $(\varepsilon = 3, \delta = 1/2N)$. The test accuracy for these models are in Figure 4. Epochs are normalized by the size of the original CIFAR-10 dataset, so training for $T$ epochs corresponds to training on $T \cdot 50,000$ examples. Note that we report the *base* learning rate, before scaling by a factor of $8192/512$.

| | ScatterNet+Linear | | ScatterNet+CNN | | CNN | |
|---|---|---|---|---|---|---|
| $N$ | Epochs $T$ | Learning rate $\eta$ | Epochs $T$ | Learning rate $\eta$ | Epochs $T$ | Learning rate $\eta$ |
| 10K | 30 | $1/8$ | 60 | $1/8$ | 30 | $1/8$ |
| 25K | 30 | $1/4$ | 60 | $1/8$ | 60 | $1/8$ |
| 50K | 60 | $1/4$ | 60 | $1/4$ | 60 | $1/4$ |
| 100K | 60 | $1/2$ | 120 | $1/4$ | 120 | $1/4$ |
| 250K | 120 | $1/2$ | 120 | 1 | 120 | 1 |
| 550K | 120 | 1 | 120 | 1 | 120 | 1 |

## C.8 EVALUATION OF PRIVATE TRANSFER LEARNING

For the transfer learning experiments in Figure 5, we use a ResNeXt-29 model pre-trained on CIFAR-100,[9] and a ResNet-50 model trained on unlabeled ImageNet (Deng et al., 2009) using SimCLRv2 (Chen et al., 2020b).[10]

To train private linear classifiers on CIFAR-10, we first extract features from the penultimate layer of the above pre-trained models. For the ResNeXt model, we obtain features of dimension 1024, and for

---

[8]The full Tiny Images dataset was recently withdrawn by its curators, following the discovery of a large number of offensive class labels (Prabhu & Birhane, 2020). The subset collected by Carmon et al. (2019) contains images that most closely match the original CIFAR-10 labels, and is thus unlikely to contain offensive content.

[9]https://github.com/bearpaw/pytorch-classification.

[10]https://github.com/google-research/simclr.

the SimCLRv2 ResNet, we obtain features of dimension 4096. We then use DP-SGD with a similar setup as for the linear ScatterNet classifiers, except that we do not normalize the extracted features. We also target a tighter privacy budget of $(\varepsilon = 2, \delta = 10^{-5})$. We then run a hyper-parameter search as listed below in Table 17. Figure 5 shows the best test accuracy achieved for each DP budget, averaged across five runs. We further report the set of hyper-parameters that resulted in the maximal accuracy for the targeted privacy budget of $(\varepsilon = 2, \delta = 10^{-5})$.

Table 17: Hyper-parameters for the evaluation of private transfer learning from CIFAR-100 (using a ResNeXt model) and from unlabeled ImageNet (using a SimCLR v2 model) in Section 5.2.

| Parameter | Values | Best for ResNeXt | Best for SimCLRv2 |
|---|---|---|---|
| DP guarantee $(\varepsilon, \delta)$ | $(2, 10^{-5})$ | - | - |
| Gradient norm $C$ | 0.1 | - | - |
| Momentum | 0.9 | - | - |
| Batch size $B$ | $\{512, 1024, \ldots, 16384\}$ | 2048 | 1024 |
| Learning rate $\eta$ | $\{1/2, 1, 2, 4\} \cdot B/512$ | $2 \cdot 2048/512$ | $2 \cdot 1024/512$ |
| Epochs $T$ | $\{15, 25, 40\}$ | 40 | 40 |

# D  ADDITIONAL EXPERIMENTS AND FIGURES

## D.1  ON THE EFFECT OF BATCH SIZES IN DP-SGD

In this section, we revisit the question of the selection of an optimal batch size for DP-SGD. In their seminal work, Abadi et al. (2016) already investigated this question, and noted that the choice of batch size can have a large influence on the privacy-utility tradeoff. They empirically found that for a dataset of size $N$, a batch size of size approximately $\sqrt{N}$ produced the best results. However, their experiments measured the effect of the batch size while keeping other parameters, including the noise multiplier $\sigma$ and the learning rate $\eta$, *fixed*.

When training without privacy, it has been shown empirically that the choice of batch size has little effect on the convergence rate of SGD, *as long as the learning rate $\eta$ is scaled linearly with the batch size* (Goyal et al., 2017). Hereafter, we argue formally and demonstrate empirically that if we use a linear learning rate scaling, and fix the number of training epochs $T$ for a target privacy budget $\varepsilon$, then the choice of batch size also has a minimal influence on the performance of DP-SGD.

We first consider the effect of the sampling rate $B/N$ on the noise scale $\sigma$ required to attain a fixed privacy budget of $\varepsilon$ after $T$ epochs. There is no known closed form expression for $\sigma$, so it is usually estimated numerically. We empirically establish the following claim, and verify numerically that it holds for our setting in Figure 6:

**Claim D.1.** *Given a fixed DP budget $(\varepsilon, \delta)$ to be reached after $T$ epochs, the noise scale $\sigma$ as a function of the sampling rate $B/N$ is given by $\sigma(B/N) \approx c \cdot \sqrt{B/N}$, for some constant $c \geq 0$.*

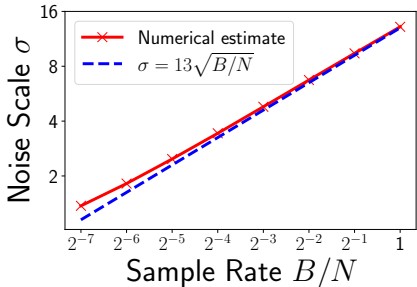

Figure 6: Noise scale $\sigma$ for DP-SGD that results in a privacy guarantee of $(\varepsilon = 3, \delta = 10^{-5})$ after 60 training epochs, for different batch sampling rates $B/N$.

Given this relation between batch size and noise scale, we proceed with a similar analysis as in (Goyal et al., 2017), for the case of DP-SGD. Given some initial weight $\boldsymbol{\theta}_t$, performing $k$ steps of DP-SGD with clipping norm $C = 1$, batch size $B$, learning rate $\eta$ and noise scale $\sigma$ yields:

$$\boldsymbol{\theta}_{t+k} = \boldsymbol{\theta}_t - \eta \sum_{j<k} \frac{1}{B} \Big( \sum_{\boldsymbol{x} \in \boldsymbol{B}_{t+j}} \tilde{\boldsymbol{g}}_{t+j}(\boldsymbol{x}) + \mathcal{N}(0, \sigma^2 \boldsymbol{I}) \Big)$$

$$= \Big( \boldsymbol{\theta}_t - \eta \frac{1}{B} \sum_{j<k} \sum_{\boldsymbol{x} \in \boldsymbol{B}_{t+j}} \tilde{\boldsymbol{g}}_{t+j}(\boldsymbol{x}) \Big) + \mathcal{N}\Big(0, \frac{k\eta^2\sigma^2}{B^2} \boldsymbol{I}\Big)$$

If we instead take a single step of DP-SGD with larger batch size $kB$, a linearly scaled learning rate of $k\eta$, and an adjusted noise scale $\tilde{\sigma} = \sqrt{k}\sigma$ (by Claim D.1), we get:[11]

$$\boldsymbol{\theta}_{t+1} = \boldsymbol{\theta}_t - k\eta \frac{1}{kB} \Big( \sum_{j<k} \sum_{\boldsymbol{x} \in \boldsymbol{B}_{t+j}} \tilde{\boldsymbol{g}}_t(\boldsymbol{x}) + \mathcal{N}(0, \tilde{\sigma}^2 \boldsymbol{I}) \Big)$$

$$= \Big( \boldsymbol{\theta}_t - \eta \frac{1}{B} \sum_{j<k} \sum_{\boldsymbol{x} \in \boldsymbol{B}_{t+j}} \tilde{\boldsymbol{g}}_t(\boldsymbol{x}) \Big) + \mathcal{N}\Big(0, \frac{k\eta^2\sigma^2}{B^2} \boldsymbol{I}\Big)$$

Thus, we find that the total noise in both updates is identical. Under the same heuristic assumption as in (Goyal et al., 2017) that $\tilde{\boldsymbol{g}}_t(\boldsymbol{x}) \approx \tilde{\boldsymbol{g}}_{t+j}(\boldsymbol{x})$ for all $j < k$, the two DP-SGD updates above are thus similar. This analysis suggests that as in the non-private case (Goyal et al., 2017), increasing the batch size and linearly scaling the learning rate should have only a small effect on a model's learning curve.

We now verify this claim empirically. We follow the experimental setup in Section 3, and set a privacy budget of $(\varepsilon = 3, \delta = 10^{-5})$ to be reached after a fixed number of epochs $T$. For different choices of batch size $B$, we numerically compute the noise scale $\sigma$ that fits this "privacy schedule". For the initial batch size of $B_0 = 512$, we select a base learning rate $\eta$ that maximizes test accuracy at epoch $T$. As we increase the batch size to $B = kB_0$, we linearly scale the learning rate to $k\eta$. The concrete parameters are given below:

Table 18: Hyper-parameters for comparing the convergence rate of DP-SGD with different batch sizes in Figure 7.

|  | Epochs $T$ | Batch size $B$ | Learning rate $\eta$ |
|---|---|---|---|
| MNIST | 40 | $\{512, 1024, 2048, 4096\}$ | $1/2 \cdot B/512$ |
| Fashion-MNIST | 40 | $\{512, 1024, 2048, 4096\}$ | $1 \cdot B/512$ |
| CIFAR-10 | 60 | $\{512, 1024, 2048, 4096\}$ | $1/4 \cdot B/512$ |

As we can see in Figure 7, the training curves for CNNs trained with DP-SGD are indeed near identical across a variety of batch sizes.

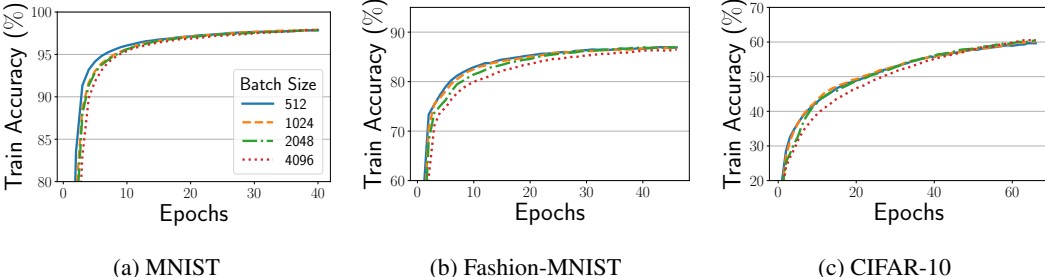

      (a) MNIST            (b) Fashion-MNIST           (c) CIFAR-10

Figure 7: Convergence rate of DP-SGD for different batch sizes, with a fixed targeted privacy budget of $(\varepsilon = 3, \delta = 10^{-5})$ after $T = 40$ or $T = 60$ epochs, and linear scaling of the learning rate $\eta \cdot B/512$.

---

[11] We make a small simplification to our analysis here and assume that one batch of DP-SGD sampled with selection probability $kB/N$ is identical to $k$ batches sampled with selection probability $B/N$.

## D.2 ANALYSIS OF HYPER-PARAMETERS

To understand the effect of varying the different hyper-parameters of DP-SGD, Figure 8 shows the median and maximum model performance for different choices of a single parameter. The median and maximum are computed over all choices for the other hyper-parameters in Table 12. As we can see, the maximal achievable test accuracy is remarkably stable when fixing one of the algorithm's hyper-parameters, with the exception of overly large batch sizes or overly low learning rates for end-to-end CNNs.

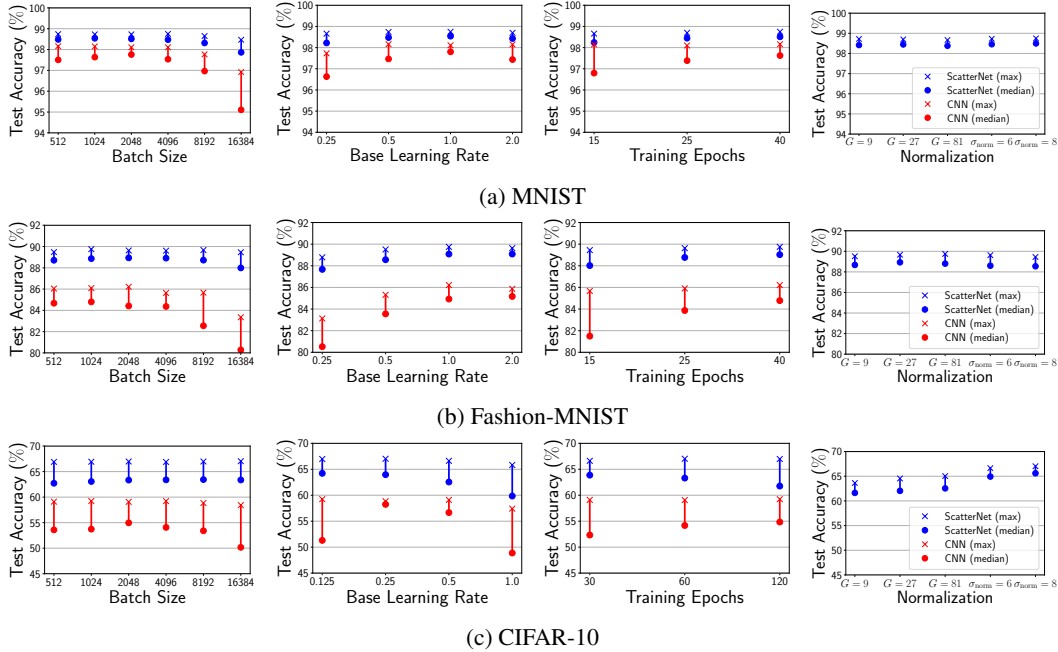

(a) MNIST

(b) Fashion-MNIST

(c) CIFAR-10

Figure 8: Median and maximum test accuracy of linear ScatterNet classifiers and end-to-end CNNs when we fix one hyper-parameter in Table 12 and run a grid-search over all others (for a privacy budget of $(\varepsilon = 3, \delta = 10^{-5})$).

## D.3 COMPARING DP-SGD AND PRIVACY AMPLIFICATION BY ITERATION

While DP-SGD is the algorithm of choice for differentially private *non-convex* learning, it is unclear why it should be the best choice for learning private *linear models*. Indeed, starting with the work of Chaudhuri et al. (2011), there have been many other proposals of algorithms for private convex optimization with provable utility guarantees, e.g., (Bassily et al., 2014; Kifer et al., 2012; Feldman et al., 2018). Yet, Yu et al. (2019a) show that DP-SGD can achieve higher utility than many of these approaches, both asymptotically and empirically.

Here, we take a closer look at the "Privacy Amplification by Iteration" work of (Feldman et al., 2018). Feldman et al. (2018) observe that DP-SGD guarantees differential privacy for *every* gradient update step. Under the assumption that intermediate model updates can be hidden from the adversary, they propose a different analysis of DP-SGD for convex optimization problems that has a number of conceptual advantages. First, the algorithm of Feldman et al. (2018) does not require the training indices selected for each batch $B_t$ do be hidden from the adversary. Second, their approach can support much smaller privacy budgets than DP-SGD.

However, we show that these benefits come at a cost in practice: for the range of privacy budgets we consider in this work, DP-SGD requires adding *less* noise than Privacy Amplification by Iteration (PAI). To compare the two approaches, we proceed as follows: We analytically compute the noise scale $\sigma$ that results in a privacy guarantee of $(\varepsilon, \delta = 10^{-5})$ after 10 training epochs with a batch sampling rate of $512/50000$.[12] Figure 9 shows that DP-SGD requires adding less noise, except for large

---

[12]The guarantees of Privacy Amplification by Iteration apply unevenly to the elements of the training data. We choose the noise scale so that at least 99% of the data elements enjoy $(\varepsilon, \delta)$-DP.

privacy budgets ($\varepsilon > 40$), or very small ones ($\varepsilon < 0.2$). In the latter case, both algorithms require adding excessively large amounts of noise. We observe a qualitatively similar behavior for other sampling rates.

For completeness, we evaluate the PAI algorithm of Feldman et al. (2018) for training linear ScatterNet classifiers on CIFAR-10. We evaluate a broader range of hyper-parameters, including different clipping thresholds $C \in \{0.1, 1, 10\}$ (PAI clips the data rather than the gradients), a wider range of batch sizes $B \in \{32, 64, \ldots, 2048\}$, and a wider range of base learning rates $\eta \in \{2^{-3}, 2^{-2}, \ldots, 2^3\}$. We find that for privacy budgets $1 \leq \varepsilon \leq 3$, the optimal hyper-parameters for PAI and DP-SGD are similar, but the analysis of PAI requires a larger noise scale $\sigma$. As a result, PAI performs worse than DP-SGD, as shown in Figure 10.

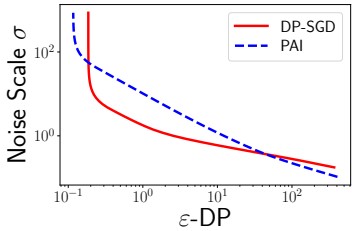

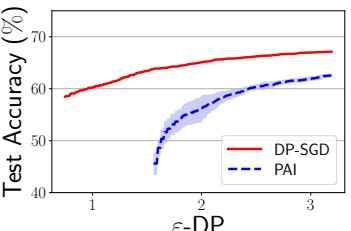

Figure 9: Gradient noise scale $\sigma$ required for a privacy guarantee of $(\varepsilon, \delta = 10^{-5})$ after 10 training epochs with batch sampling rate $^{512}/_{50000}$. Privacy Amplification by Iteration (PAI) (Feldman et al., 2018) requires less noise than DP-SGD only for very small or very large privacy budgets.

Figure 10: Comparison of DP-SGD (Abadi et al., 2016) and Privacy Amplification by Iteration (PAI) (Feldman et al., 2018) for training a private linear ScatterNet classifier on CIFAR-10. Shows the maximum accuracy achieved for each privacy budget, averaged over five runs.

### D.4 DP-SGD WITH POISSON SAMPLING

The analysis of DP-SGD (Abadi et al., 2016; Mironov et al., 2019) assumes that each batch $B_t$ is created by independently selecting each training sample with probability $^B/_N$. This is in contrast to typical implementations of SGD, where the training data is randomly shuffled once per epoch, and divided into successive batches of size *exactly* $B$. The latter "random shuffle" approach has been used in most implementations of DP-SGD (e.g., (tensorflow/privacy, 2019; pytorch/opacus, 2020)) as well as in prior work (e.g., (Abadi et al., 2016; Papernot et al., 2020b)), with the (implicit) assumption that this difference in batch sampling strategies will not affect model performance. We verify that this assumption is indeed valid in our setting. We re-train the linear ScatterNet and end-to-end CNN models that achieved the highest accuracy for a DP budget of $(\varepsilon = 3, \delta = 10^{-5})$ (with the hyper-parameters detailed in Table 13), using the correct "Poisson sampling" strategy. The test accuracy of these models (averaged over five runs) are shown in Table 19. For all datasets and models, the two sampling schemes achieve similar accuracy when averaged over five runs.

Table 19: Comparison of DP-SGD with two different batch sampling schemes: (1) *Poisson sampling*, where a batch is formed by selecting each data point independently with probability $^B/_N$; (2) *Random shuffle*, where the training set is randomly shuffled at the beginning of each epoch, and split into consecutive batches of size $B$. For both sampling schemes, we report the best test accuracy (in %) at a DP budget of $(\varepsilon = 3, \delta = 10^{-5})$, with means and standard deviations over five runs.

| | ScatterNet | | CNN | |
| --- | --- | --- | --- | --- |
| Dataset | Poisson Sampling | Random Shuffle | Poisson Sampling | Random Shuffle |
| MNIST | $98.6 \pm 0.1$ | $98.7 \pm 0.0$ | $98.0 \pm 0.1$ | $98.1 \pm 0.0$ |
| Fashion-MNIST | $89.6 \pm 0.1$ | $89.7 \pm 0.0$ | $86.1 \pm 0.2$ | $86.0 \pm 0.1$ |
| CIFAR-10 | $66.8 \pm 0.2$ | $67.0 \pm 0.0$ | $59.0 \pm 0.4$ | $59.2 \pm 0.1$ |

### D.5 EXPERIMENTS WITH SMALLER END-TO-END CNN MODEL ON CIFAR-10

In Section 4, we investigate whether the dimensionality of different classifiers has a noticeable impact on their privacy-utility tradeoffs. To this end, we repeat the CIFAR-10 experiments from

Section 3 with a smaller end-to-end CNN architecture. Specifically, we take the end-to-end CNN architecture from Table 7 and reduce the number of filters in each convolutional layer by a factor of two and remove the last convolutional layer). This results in a CNN model with a comparable number of trainable parameters as the linear ScatterNet classifier (see Table 4). In Table 20, we compare the privacy-utility of this smaller CNN models with the original larger CNN model evaluated in Section 3. While the change of model architecture does affect the model accuracy, the effect is minor, and the accuracy remains far below that of the ScatterNet classifiers with a comparable number of parameters.

Table 20: Best test accuracy (in %) for two different model sizes on CIFAR-10 for a DP budget of ($\varepsilon = 3, \delta = 10^{-5}$). We compare two variants of the end-to-end CNN architecture from Table 7, with respectively 551K and 168K parameters. Average and standard deviation computed over five runs.

| Model | Parameters | Accuracy |
|---|---|---|
| CNN | 168K | $60.7 \pm 0.3$ |
| | 551K | $59.2 \pm 0.1$ |

### D.6 MODEL CONVERGENCE SPEED ON MNIST AND FASHION-MNIST

We run the same experiment as in Figure 3 for MNIST and Fashion-MNIST, to compare the convergence rate of different classifiers with and without privacy, for different learning rates. The experimental setup is described in Appendix C.6. Figure 11 shows qualitatively similar results as Figure 3: with a high learning rate, all models converge quickly when trained without gradient noise, but the addition of noise is detrimental to the learning process. In contrast, with a much lower learning rate the training curves for DP-SGD are nearly identical, whether we add noise or not. In this regime, the ScatterNet classifiers converge significantly faster than end-to-end CNNs when trained *without* privacy.

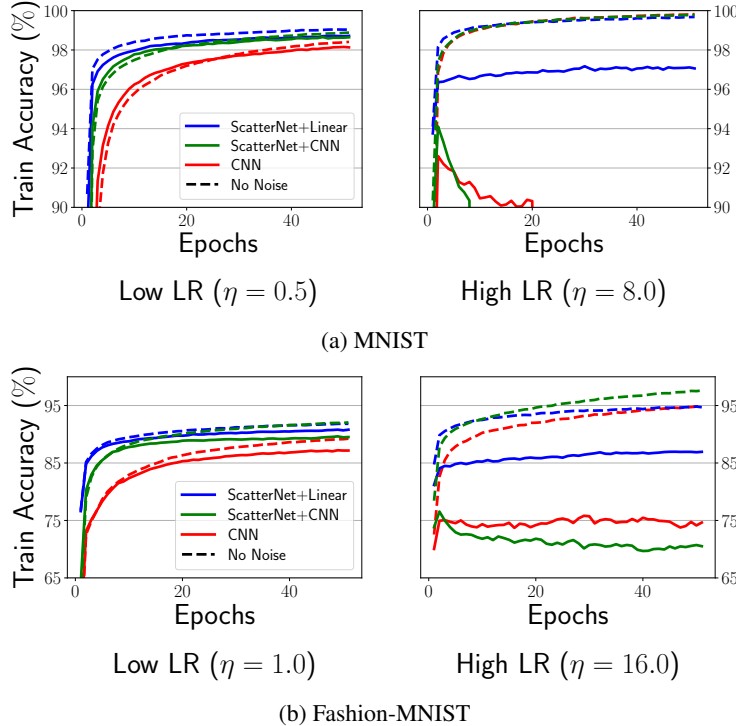

Figure 11: Comparison of convergence rates of linear classifiers fine-tuned on ScatterNet features, CNNs fine-tuned on ScatterNet features), and end-to-end CNNs with and without noise addition in DP-SGD. (Left): low learning rate. (Right): high learning rate. See Figure 3 for results on CIFAR-10.

