# OpenReview forum: "Differentially Private Learning Needs Better Features (or Much More Data)"
_ICLR.cc/2021/Conference — ICLR 2021 Spotlight_

### Official Review · AnonReviewer1 · 2020-10-24

**Rating:** 6
**Confidence:** 5

**Review:**

The paper presents an analysis of differential privacy in machine learning, with a focus on neural networks trained via differentially private stochastic gradient descent (DPSGD). The main focus and the message in the paper is that the handcrafted features work better compared to learned features during training of NNs and having more training data results in better outcomes (i.e. a better privacy-utility trade-off).

Starting with the latter, this is apparent from the noise formulation in DPSGD, where the noise is reduced via sampling probability, which decreases as the data size grows. Hence, I do not consider this as a new insight or a contribution. Unless, I have misunderstood something, in which case, please do explain.

For the former, as the final model used (Table 3 and Figure 1) is a linear classifier, it outperforming an end-to-end CNN based model is intuitive, as it has far fewer number of parameters (which improve the noise scale, due to smaller gradient norm). This is slightly touched upon in subsection "Training CNNs on handcrafted features", where the comparison is made using CNNs on the handcrafted features, however there are no detailed results presented in the paper, I would have liked to see a similar table and figures as earlier.

The presentation of results (Table 3) is a bit strange. I would have further liked to see the comparison of both models on the *same* set of hyperparameters. Also, instead of stating that hyperparameter search's privacy budget was not accounted for as in prior works, it would have been nice to see some analysis, such as the section D(Appendix) in Abadi et al.

---

> ### Author Response · Authors · 2020-11-17
> **Response to Reviewer 1**
>
> We thank the reviewer for their insightful comments.
>
> **1)    Q:** *...having more training data results in better outcomes [...] I do not consider this as a new insight or a contribution. Unless, I have misunderstood something, in which case, please do explain.*
>
> **A:** As many reviewers raised a similar concern about expectedness of our results when training with more data, we provide a detailed response to these concerns in a meta-comment above.
>
> *In summary, while the tradeoff we show is indeed expected qualitatively, we are interested in a quantitative assessment: how much more private data is needed for private end-to-end deep learning to outperform handcrafted features?*
>
> Our result shows that improving private deep learning simply by collecting more data might be very expensive for canonical vision tasks like CIFAR-10: we need about one order of magnitude more data before the end-to-end CNN outperforms our handcrafted baselines. This motivates the design of better private learning algorithms in low data regimes.
>
> **2)    Q:** *as the final model used (Table 3 and Figure 1) is a linear classifier, it outperforming an end-to-end CNN based model is intuitive, as it has far fewer number of parameters*
>
> **A:** **The claim that our linear classifiers have fewer parameters than end-to-end CNNs is incorrect!** This is the point of our analysis in Section 4: “Smaller models are not easier to train privately.”
>
> We find that ScatterNet models outperform end-to-end CNNs *despite having more trainable parameters!* As the reviewer correctly notes, the noise analysis of DP-SGD would suggest the opposite, **so this result is surprising.**
> We have clarified this in our introduction.
>
> As we show in Section 4, handcrafted features give rise to an “easier” learning task, where convergence occurs much faster *despite the higher noise.*
>
> **3)    Q:**  *there are no detailed results presented in the paper, I would have liked to see a similar table and figures as earlier*
>
> **A:** It is unclear to us what the reviewer means by a lack of “detailed results” for CNNs trained on ScatterNet features. Figure 2 is analogous to the earlier Figure 1 and directly compares the ScatterNet+linear, ScatterNet+CNN and end-to-end CNN models for all privacy budgets we consider.
> We have also added results with ScatterNet+CNN models to Table 1 and Table 3, which in particular includes some improved results for the ScatterNet+CNN model on CIFAR-10, following a suggestion of reviewer 3.
> Are there other results that we could add to the paper to better compare the ScatterNet and end-to-end results?
>
> **4)    Q:** *The presentation of results (Table 3) is a bit strange. I would have further liked to see the comparison of both models on the same set of hyperparameters*
>
> **A:** We are not sure what the reviewer means by a “comparison of both models on the same set of hyperparameters”. It seems to us that this is exactly what is shown in Figure 1, which shows the best performance achieved by both linear ScattterNet models and end-to-end CNNs for each privacy budget, across the entire hyper-parameter set.
>
> If the reviewer meant comparing these models for a *single* assignment of hyper-parameters, this is somewhat tricky to do in a fair way. It is unclear how to choose those hyper-parameters so as not to favor one of the two models.
> We note that for CIFAR-10 this issue is moot anyhow: *the best hyper-parameter assignment for end-to-end CNNs is outperformed by the worst assignment of hyper-parameters for linear ScatterNet models (Table 3).*
>
> **4)    Q:** *instead of stating that hyper-parameter search's privacy budget was not accounted for as in prior works, it would have been nice to see some analysis, such as the section D(Appendix) in Abadi et al.*
>
> **A:** We can definitely perform an analysis of the cost of hyper-parameter search as in Abadi et al. Ultimately though, none of the prior works that we compare against in Table 1 do this, so it is hard to perform a fair comparison that accounts for this cost. If we solely consider our comparison of end-to-end CNNs and ScatterNet models, then the hyper-parameter sets are identical so the analysis of Abadi et al. would yield the same cost in both cases. Moreover, as we show in Table 2 and Appendices D.1 and D.2, our hyper-parameter search was anyhow somewhat “excessive”, especially for linear ScatterNets. We would obtain similar results by considering a much smaller set of parameters (e.g., we find that contrary to what was claimed in prior work, the choice of batch size is not very important in DP-SGD, as long as the learning rate is set adequately).

---

> > ### Comment · AnonReviewer1 · 2020-11-22
> > **Response to authors**
> >
> > Thank you. I very much appreciate the detailed response.
> >
> > Starting with Q2, I still do not understand how the linear classifier has more parameters than a deep CNN. Please correct me if I am wrong: Linear model = logistic regression(or a single layer NN) on top of extracted features via scatter net? if that is the case, then the number of parameters should be smaller, no? i.e. a single layer NN compared to multilayer NN. Or am I missing any obvious details?
> >
> > Q4: Thanks, that remark makes it clear. I meant was to use the set of hyperparameters from a previous study for competitors and run your model with a similar set of hyperparameters.
> >
> > Q5: Thanks. Choice of batch size is not very important in DPSGD: I disagree with the statement as we can see from theorem 1 of Abadi et al. that the batch size is directly proportional to the noise addition. Although this is countered by the summation step (before adding noise), but making such a blanket statement is not correct as in practice, we see a significant difference, given everything else is fixed.

---

> > > ### Author Response · Authors · 2020-11-22
> > > **Response**
> > >
> > > Thank you for the follow-up to your review and for the clarifications.
> > >
> > > **Q2:** There are three factors that explain this fact: (1) the ScatterNet transform is slightly expansive; (2) Convolutions have fewer parameters than dense layers due to weight sharing; (3) Deep networks typically downsample the input (e.g., via maxpooling or convolutions of stride > 1) before the final dense layers.
> > >
> > > First, note that CNNs having fewer parameters than linear models is very common. E.g., on ImageNet with images of 224x224 pixels, a Logistic regression model would have approximately 150M parameters (224\*224\*3\*1000 + 1000). In contrast, a ResNet-50 has only about 25M parameters.
> > >
> > > Let us illustrate with concrete numbers for the models we use for MNIST and Fashion-MNIST:
> > > - The input size is 28x28x1=784. A linear model on top of pixels (which we don't consider in our paper) would have 7850 parameters.
> > > - The ScatterNet transform outputs features of dimension 7x7x81 (i.e., the transform combines spatial downsampling and channel upsampling). **The linear ScatterNet classifiers have 39700 parameters** (7\*7\*81\*10 + 10).
> > > - The end-to-end CNN has two convolutional layers, each followed by a maxpool layer, and then two dense layers. Due to the weight-sharing of convolutions and the spatial downsampling of both the convolutions and maxpool layers, this model only has **26010 parameters.**
> > > -  The ScatterNet+CNN model also has maxpool layers that downsample the features before the dense layers. This model has **33066 parameters.**
> > >
> > > If we had used deep *dense* models, then those would indeed have more parameters than the linear models.
> > >
> > > **Q4:** Thank you for this clarification. We agree that re-using the same set of hyperparameters as in a prior work would be a good approach. Unfortunately, prior work rarely provides these details. E.g., the best source of comparison would be the work of Papernot et al. which proposed the CNNs we use, but their paper does not list the hyperparameters considered. The work of Abadi et al. gave more explicit hyperparameters, but their setup differs too significantly from ours to allow for a direct comparison (Abadi et al. use private PCA before training MNIST models, and they use transfer-learning for CIFAR-10).
> > >
> > > We hope that our approach of listing all hyperparameter searches performed will encourage future work in this area to also provide these important details.
> > >
> > > **Q5:** Apologies, we should clarify what we meant by this. We agree that, all other parameters being fixed, the choice of batch-size is important for DP-SGD. *But the same is also true for regular SGD!*
> > >
> > > Indeed, there is a large literature on batch size scaling in the non-private case, that shows that when multiplying the batch size by a factor k, the learning rate should also be multiplied by k to keep the performance constant. As we show in Appendix D.1, the same simple linear scaling rule also holds for DP-SGD.
> > >
> > > In Appendix D.1, we further show formally that if we scale both the learning rate and the batch size, and adjust the noise level according to the analysis of Abadi et al., then the amount of noise that DP-SGD injects per epoch remains the same. Our experimental results (Figure 7) confirm this analysis: model convergence stays the same as we increase the batch size from 512 to 4096 with a linearly scaled learning rate (and all other parameters fixed).
> > >
> > > So a more precise statement would be: *with all other parameters fixed, tuning the batch size of DP-SGD is similar to tuning its learning rate*. As tuning the learning rate is much more common with regular SGD, we recommend to do the same with DP-SGD.

---

> > > > ### Comment · AnonReviewer1 · 2020-11-23
> > > > **Response**
> > > >
> > > > Q2. Thank you for the explanation. I had missed the scatternet transform's property of "increasing dimensionality".  This makes sense. I was under the impression that automated feature extraction would result in lower dimensionality than the original dimensions.
> > > >
> > > > Do you have any comment on the applicability of the proposed tricks on tabular data? As I assume the inherent differences between image and tabular data would result in significantly different feature extraction and/or the subspace on which they reside.
> > > >
> > > > Q4. I agree.
> > > >
> > > > Q5. Yes, that is much better. Can you please provide the details on the computing infrastructure used for the experiments? As I understand, increasing batch size takes a heavy toll on DPSGD's performance.

---

> > > > > ### Author Response · Authors · 2020-11-23
> > > > > **Great questions**
> > > > >
> > > > > Thanks for the response, we are glad that we could clarify these points.
> > > > >
> > > > > And thank you for raising these additional insightful questions. We will also clarify these points in our writeup.
> > > > >
> > > > > **Q2:** The ScatterNet transform we use was indeed designed for images, as it aims to capture invariants that are specific to natural image data (e.g., invariance under small rotations and translations).
> > > > >
> > > > > Alternative transforms have been proposed for other domains such as speech (https://arxiv.org/abs/1304.6763) and chemistry (https://math.msu.edu/user_content/docs/m10754520190706225610788.pdf).
> > > > >
> > > > > For domains with tabular data, the Scattering transform is probably not directly applicable. Nevertheless, whenever we have priors on the data domain, domain-specific handcrafted features may be available, and *our thesis is that these handcrafted features are beneficial for private learning*.
> > > > >
> > > > > For example, we are very interested in exploring extensions of our work to natural language processing, a domain with a rich history of task-specific handcrafted features, which may also prove useful to improve private deep learning.
> > > > >
> > > > > **Q5:** All our experiments are performed on a single TITAN xp GPU with 12GB of memory.
> > > > >
> > > > > As DP-SGD requires computing per-example gradients, we indeed cannot fit all gradients in memory at the same time.
> > > > > To solve this, we make use of the "virtual step" feature in opacus (https://github.com/pytorch/opacus). Basically, we split each batch into "mini batches" that fit in memory (we use mini batches of size 256 in all our experiments). For each mini-batch, we compute per-example gradients, clip them, and sum them up. Once we've processed an entire batch, we add the noise and take the actual update step. Note that this "caching" approach has no influence on the privacy analysis of DP-SGD.
> > > > >
> > > > > The process is very similar to how standard SGD can be scaled to large batch sizes that do not fit in GPU memory.
> > > > >
> > > > > We have uploaded code to reproduce all our experiments.

---

> > > > > > ### Comment · AnonReviewer1 · 2020-11-23
> > > > > > **Response**
> > > > > >
> > > > > > Thank you for all the answers. I have updated my score accordingly.

---

> > > > > > > ### Author Response · Authors · 2020-11-23
> > > > > > > **Response**
> > > > > > >
> > > > > > > Thank you for your appreciation, and for the fruitful discussion which will help us improve the presentation of our results.

---

### Official Review · AnonReviewer2 · 2020-10-27
**Nice experiments**

**Rating:** 7
**Confidence:** 3

**Review:**

The paper shows that linear model on top of ScatterNet can outperform CNN for DPSGD training on a few generic image classification tasks. It analyzed the results, provides hypotheses to explain it, and concludes that more data / better feature is needed for DPSGD training.

People have been searching for good models for DPSGD training, so it is nice to see a new baseline (ScatterNet + linear model) that is simple but performs better. This can be pretty valuable for researchers and practitioners in the field.
The paper also did some quite interesting experiments to explain the advantage of ScatterNet + linear model and to suggest directions to improve DPSGD training. The experiments and discussions on the learning rate are quite interesting and inspiring to me. However, I feel like the results for having more data / transfer learning is not so surprising, though the experiments with models different from previous work are valuable.

More detailed comments:
- In Sec 4 "smaller models are not easier to train privately", you mentioned that the CNN is smaller than the linear model, so dimensionality is not an explanation for ScatterNet + linear's better performance. But I guess convex model (or maybe shallow model) might have some fundamental difference from nonconvex (or maybe deeper model). Maybe you could try something deeper than linear but shallower than the CNN to see if there is a sweet spot in between.
- In Sec 4 "Models with handcrafted features converge faster without privacy", I guess the results can be explained by the fact that simpler model (linear) has a lower capacity than more complicated model (CNN) so requires less training time even with lower learning rate. So maybe again it would worth trying something in between linear and CNN.
- In Sec 5.2, you showed results that are much better than previous results with transfer learning. It seems like the main difference is the model architecture. Is that the case? Do you have any comment on that?

The presentation is clear in general. I would love to see more details about ScatterNet as that is the important component of the proposed method.

---

> ### Author Response · Authors · 2020-11-17
> **Response to Reviewer 2**
>
> We thank the reviewer for their insightful comments.
>
> **1)    Q:** *However, I feel like the results for having more data / transfer learning is not so surprising, though the experiments with models different from previous work are valuable.*
>
> **A:** As many reviewers raised a similar concern about expectedness of our results when training with more data or with transfer learning, we provide a detailed response to these concerns in a meta-comment above.
>
> In summary, while the data tradeoff we show is indeed expected *qualitatively*, we are interested in a *quantitative* assessment: how much more private data is needed for private end-to-end deep learning to work?
> Similarly, our experiments with transfer learning aim to rectify confusion in the literature about the quantitative limitations of DP in this setting, which seem largely due to prior work only evaluating private transfer learning with very weak public source models (as the reviewer notes, the only difference in our approach is the use of a state-of-the-art source model. It is unclear to us why prior work has not followed this approach).
>
> **2)    Q:** *Maybe you could try something deeper than linear but shallower than the CNN to see if there is a sweet spot in between.*
>
> **A:** We agree with the reviewer that the non-convexity or larger capacity of neural networks are likely a cause for their slower convergence (with or without ScatterNet features). Following the reviewers' great suggestion, we experimented with shallower CNN architectures on MNIST, Fashion-MNIST and CIFAR-10, by varying the number of convolutional layers and dense layers.
> We observe no improvement for the end-to-end CNNs (this is not very surprising, as the original CNNs we use were the result of an architecture search tailored to DP-SGD performed by Papernot et al.)
> **However, for the Scat+CNN model on CIFAR-10, we do see some improvements with shallower architectures!**
>
> Specifically, our best Scat+CNN model slightly outperforms our best linear model on CIFAR-10 (we get 69% accuracy at eps=3).
> This is exciting! We have added these improved results to our submission, and we plan to perform a more thorough architecture search for ScatterNet models.
> For MNIST and Fashion-MNIST, we could not improve the Scat+CNN results further by varying the model depth. The architectures we use are already very shallow (2 convolutions followed by 2 dense layers) and further reducing the model size reduces private and non-private accuracy alike.
>
> The main reason for this discrepancy between (Fashion)-MNIST and CIFAR-10 is that on the simpler MNIST tasks, the linear ScatterNet model achieves near optimal accuracy in the non-private setting. Therefore, deeper architectures have little room for improvement, and since they converge more slowly, they perform worse in the private setting. On CIFAR-10 however, the linear ScatterNet model achieves about 71% non-private accuracy, which is far from state-of-the-art. Our shallow Scat+CNN model achieves about 74% non-private accuracy, so even though it converges a bit slower, the extra base accuracy is sufficient to slightly outperform the linear model in the private setting. The end-to-end CNN is at the other end of this spectrum: it achieves over 80% accuracy in the non-private setting, but converges too slowly to compete in the private setting. These results are included in our updated submission.
>
> Our main conclusion remains that handcrafted features are highly beneficial for private learning in low data regimes.
>
> **3)    Q:** *In Sec 5.2, you showed results that are much better than previous results with transfer learning. It seems like the main difference is the model architecture. Is that the case? Do you have any comment on that?*
>
> **A:** This indeed seems to be the main difference (e.g., the model used by Papernot et al. achieves 75% accuracy in the non-private setting). Our results thus show that as in the non-private case, the performance of private transfer learning is highly dependent on the choice of a good source model. A thorough analysis of differentially private transfer learning is out of scope of our paper, but our results provide strong baselines to inform such a study in the future.
>
> **4)    Q:** *I would love to see more details about ScatterNet*
>
> **A:** We do provide some details on ScatterNets in Appendix C.1. Are there additional aspects of these networks that would be helpful for us to include there?

---

### Official Review · AnonReviewer3 · 2020-10-28
**New baselines for differentially private vision tasks using handcrafted feature extractors**

**Rating:** 7
**Confidence:** 4

**Review:**

This article is about a topical issue: performance degradation of of deep learning models trained with differential privacy (DP). Clipping of the gradients and addition of the noise, required to obtain DP guarantees, blur the models such that for moderate privacy guarantees (eps~7.0) CIFAR-10 test accuracy baseline is currently ~66% (Papernot et al., 2020b).

Lower bounds for this degradation have been shown theoretically (e.g. Bassily et al., 2014), and there has recently been also work on circumventing this issue, see e.g.

Kairouz, P., Ribero, M., Rush, K. and Thakurta, A., 2020. Dimension Independence in Unconstrained Private ERM via Adaptive Preconditioning. arXiv preprint arXiv:2008.06570,
Yingxue Zhou, Zhiwei Steven Wu, and Arindam Banerjee. Bypassing the ambient dimension: Private sgd with gradient subspace identification. arXiv preprint arXiv:2007.03813, 2020
(these references are not included in the paper).

Although this paper does not introduce fundamentally anything new for DP learning (DP-SGD + Rényi DP accountant for obtaining eps,delta-guarantees are used), it does clearly beat the state-of-the-art for small epsilon values (eps up to 3.0) for MNIST, Fashion-MNIST and CIFAR-10. This is obtained by using so called Scattering Networks (Oyallon and Mallat, 2015), which have the property of converging very fast without privacy. This phenomenon is transferred to DP learning and thus high accuracies for shorter DP-SGD runs (i.e. smaller epsilons) are obtained.

As expected, these 'handcrafted data-independent feature extractors' of Scatter Networks cannot beat CNN+DP-SGD when more private data is available, or when features can be extracted from public image data.

All in all, although I think the gist of the paper is simply combining these handcrafted feature extractors (ScatterNets) and DP-SGD, it does improve the baseline for DP CIFAR-10 for small / moderate eps-values (up to 3.0) and does provide new ideas / questions on how to improve DP learning (e.g. by accelerated convergence) also outside of image domain (handcrafted feature extraction for non-vision tasks).

The paper is very well written. A tiny remark:
You write "Gaussian noise of variance sigma^2 C^2 is added to the mean gradient."
Notice that sigma^2 C^2 - noise is added to the summed gradients, and sigma^2 C^2 / B^2 to the mean.

---

> ### Author Response · Authors · 2020-11-17
> **Response to Reviewer 3**
>
> We thank the reviewer for their insightful comments.
>
> **1)    Q:** *Lower bounds for this degradation have been shown theoretically (e.g. Bassily et al., 2014), and there has recently been also work on circumventing this issue*
>
> **A:** Thank you for pointing us to the works of Kairouz et al. and Banerjee et al. We added a reference to these works in our paper.
> There has indeed been a lot of work on trying to improve the performance of DP-SGD (e.g., all the works listed in Table 1). As our results show however, proposed improvements have typically been evaluated in sub-par settings, which makes it hard to convincingly argue which techniques will generalize to better models.
> For example, Banerjee et al. achieve 85% accuracy on MNIST, which is far below ours and prior results.
> We hope that our work can provide simple and strong baselines for private learning, which can be used to assess future proposed improvements to DP-SGD.
>
> **2)    Q:** *As expected, these 'handcrafted data-independent feature extractors' of Scatter Networks cannot beat CNN+DP-SGD when more private data is available, or when features can be extracted from public image data*
>
> **A:** As many reviewers raised a similar concern about expectedness of our results when training with more data or with transfer learning, we provide a detailed response to these concerns in a meta-comment above.
>
> In summary, while the data tradeoff we show is indeed expected *qualitatively*, we are interested in a *quantitative* assessment: how much more private data is needed for private end-to-end deep learning to work?
> Similarly, our experiments with transfer learning aim to rectify confusion in the literature about the quantitative limitations of DP in this setting, which seem largely due to prior work only evaluating private transfer learning with very weak public source models.

---

### Official Review · AnonReviewer4 · 2020-10-29
**Interesting experimental study of improved, differentially private image classification**

**Rating:** 7
**Confidence:** 2

**Review:**

The paper considers ways of improving private versions of SGD in the context of image classification. The main finding is that providing "hand crafted" features can significantly improve the privacy/accuracy trade-off. In some cases, even a linear model built on top of such features (like those produced by ScatterNet), can improve over differentially private SGD. A plausible explanation for this phenomenon is that extra features can reduce the number of iterations required in SGD, resulting in better privacy and/or less noise. (It is also argued that having much more data similarly improves the trade-off, but this is unsurprising and, it seems, has been observed before by McMahan et al.)

The paper is quite well-written, and I found it easy to follow even though this is not my area of expertise. I also like that it presents a number of possible directions for further improving private SGD, including transfer learning from related, public data sets, and second-order optimization.

A possible criticism is that in principle the "hand crafted" features may have been built based on empirical work on MNIST and CIFAR-10, and the same goes for the architecture choices, so in theory there could be some privacy leakage from these choices. It would have been more impressive to demonstrate effectiveness of a newer data set, not known when ScatterNet and the used CNN architectures were proposed.

Two final comments:
- "Unlearned" usually means that you have (deliberately) forgotten something, so it is not the same as "not learned".
- It would be interesting to consider the setting where just the image *label* is private. Has DP SGD been considered in that setting?

---

> ### Author Response · Authors · 2020-11-17
> **Response to Reviewer 4**
>
> We thank the reviewer for their insightful comments.
>
> **1)    Q:** *A possible criticism is that in principle the "hand crafted" features may have been built based on empirical work on MNIST and CIFAR-10*
>
> **A:** The design of ScatterNet may indeed have been partially influenced by existing vision datasets. However, the core scattering transform originates from a purely theoretical work of Mallat. The architecture proposed by Oyallon and Mallat was indeed originally evaluated on CIFAR-10, but not on MNIST or Fashion-MNIST. The latter two datasets are qualitatively very different from CIFAR-10, so this speaks to the generality of the approach.
> Note that the Fashion-MNIST dataset (2017) was indeed proposed after the ScatterNet paper (2014).
> We agree that the choice of CNN architectures might lead to some privacy leakage, but this only strengthens our main result that linear models with handcrafted features outperform deep neural nets for moderate privacy budgets.
>
>
> **2)    Q:** *It is also argued that having much more data similarly improves the trade-off, but this is unsurprising and, it seems, has been observed before by McMahan et al.*
>
> **A:** As many reviewers raised a similar concern about expectedness of our results when training with more data, we provide a detailed response to these concerns in a meta-comment above.
>
> In summary, while the tradeoff we show is indeed expected *qualitatively*, we are interested in a *quantitative* assessment: how much more private data is needed for private end-to-end deep learning to work?
>
>
> **3)    Q:** *Two final comments:*
>
> **A:**
> - Yes, thank you for pointing out this misnomer about “unlearning”
> - The label-private setting is indeed very interesting to consider. We are unaware of any work in this direction. The standard DP-SGD algorithm does not seem well suited to exploit such a weaker privacy model. Indeed, DP-SGD provides privacy at the level of gradients, which depend on both the input and label.

---

### Author Response · Authors · 2020-11-17
**Clarification of contributions and changes to submission**

The main contribution of our paper is to show that in standard data regimes, private end-to-end deep learning is outperformed by simpler (linear) models on top of handcrafted features.

Given this state of affairs, we further explore two natural avenues to bridge the gap between standard and private learning: collecting more private data, and transfer learning from public data.
The question we aim to answer here is: *what is the cost of DP’s “AlexNet” moment?*
I.e., what do we need in order for private learning to outperform handcrafted features?

Multiple reviewers have raised a similar concern, namely that it is not very surprising that these two avenues---collecting more private data, and transfer learning---lead to improved results.

**We remark that these results were not meant to be *qualitatively* surprising! Rather, our aim was to provide *quantitative* results to compare the viability of different natural approaches for improving private learning.**

These approaches, while known, haven’t been rigorously evaluated in past work.
From our experiments, we extract quantitative results that were not obvious a priori:
- Beating handcrafted features with end-to-end deep learning requires about *one order of magnitude more private training data* on CIFAR-10.
- With transfer learning from a state-of-the-art source model, differential privacy comes at only a small cost in accuracy, and significantly outperforms handcrafted features.

First, our results show that the data-complexity of private deep learning is currently much higher than for standard learning, and this motivates the design of better private learning algorithms in low data regimes (such as our approach with handcrafted features).
If the results had been quantitatively different (e.g., 10% additional data suffices to bridge the gap), our conclusions would of course be very different.

Second, our strong results with transfer learning may seem obvious in hindsight, and yet prior results in this setting have reported much worse results than ours, primarily because they built upon weak non-private transfer learning baselines.
This has led to a significant amount of confusion in the literature about the limitations of private (transfer) learning, which our paper rectifies.
As with the other results in our paper, we hope that our improved transfer learning results can act as strong baselines to compare future approaches for private learning.

**We have amended our submission to clarify the above points. We have also added improved results for the ScatterNet+CNN models on CIFAR-10, following a suggestion from reviewer 2. We find that CNNs with handcrafted features can slightly improve over the linear models. This does not change our main conclusion---i.e, handcrafted features help for private learning.**

---

### Decision · Program_Chairs · 2021-01-07
**Final Decision**

**Decision:**

Accept (Spotlight)

**Comment:**

This paper presents a very interesting investigation. While deep neural networks are typically best in non-private settings, the authors show that linear models with handcrafted features (ScatterNets) perform better in certain settings of the privacy parameter. The reviewers all found this to be important and insightful, with a thorough investigation, and I tend to agree, recommending acceptance.